# Advances in Analysis of Milk Proteases Activity at Surfaces and in a Volume by Acoustic Methods

**DOI:** 10.3390/s20195594

**Published:** 2020-09-29

**Authors:** Mark Dizon, Marek Tatarko, Tibor Hianik

**Affiliations:** 1School of Chemistry, University College Dublin, Belfield, Dublin 4, Ireland; mark.dizon@ucdconnect.ie; 2Department of Nuclear Physics and Biophysics, Faculty of Mathematics, Physics and Informatics, Comenius University, Mlynska dolina F1, 842 48 Bratislava, Slovakia; marek.tatarko@fmph.uniba.sk

**Keywords:** plasmin, trypsin, protease, casein, cleavage, acoustic sensor, thickness shear mode, quartz crystal microbalance, high-resolution ultrasonic spectroscopy

## Abstract

This review is focused on the application of surface and volume-sensitive acoustic methods for the detection of milk proteases such as trypsin and plasmin. While trypsin is an important protein of human milk, plasmin is a protease that plays an important role in the quality of bovine, sheep and goat milks. The increased activity of plasmin can cause an extensive cleavage of β-casein and, thus, affect the milk gelation and taste. The basic principles of surface-sensitive acoustic methods, as well as high-resolution ultrasonic spectroscopy (HR-US), are presented. The current state-of-the-art examples of the application of acoustic sensors for protease detection in real time are discussed. The application of the HR-US method for studying the kinetics of the enzyme reaction is demonstrated. The sensitivity of the acoustics biosensors and HR-US methods for protease detection are compared.

## 1. Introduction

Milk proteases are of high importance in the digestion of milk proteins, mostly caseins. Among them, the plasmin has a crucial role, because its activity has a substantial effect on the quality of milk and milk products. The primary role of the plasmin consists in the cleavage of fibrin in blood and, thus, prevents thrombosis. However, it is also infiltrated in milk when it is responsible for cleavage, mostly of β-casein [1]. Plasmin in blood and milk originated from its zymogen plasminogen. The cleavage of plasminogen by urokinase activators results in plasmin [2]. The activity of plasmin is regulated by a complex plasmin system that contains activators and inhibitors [3]. Due to the high temperature stability of plasmin, even after ultra-high temperature treatment (UHT) of the milk, this protease can be active. The uncontrolled activity of plasmin causes the extensive cleavage of casein on small peptide fragments that are not desirable for milk storage because of inducing its gelation and bitter taste. It is therefore important to develop methods that can control the plasmin concentration and their activity. The concentration of plasmin in raw milk is in the range of 4–12 nM [2]; therefore, the methods of plasmin determination should have sensitivity in order of 1 nM. It is also important for analysis of the kinetics of casein cleavage by proteases and their time stability.

So far, plasmin and the other protease, trypsin, present in human milk were detected by conventional methods, such as the Enzyme-linked Immunosorbent Assay (ELISA), which is currently the only one commercially available test kit for plasmin detection [4]. In this assay, the specific antibodies to plasmin are used. However, this method does not allow monitoring the kinetics of the cleavage of the caseins. Other possible assays are based on optical methods, but they are limited to only optically transparent liquids [5]. Other available methods include matrix-assisted laser desorption/ionization time of flight mass spectrometry (MALDI-TOF-MS) [6], high-performance liquid chromatography (HPLC) [7] or capillary electrophoresis [8]. However, these methods determine only the peptide fragments and not the plasmin activity. Therefore, kinetic studies of proteases activity are impossible by these methods.

It is evident that there is high demand on the development of effective methods for the detection of proteases activity. Among the effective approaches are the acoustic methods. The advantage is high sensitivity and the capability to operate in nontransparent liquids. The acoustic methods can be divided into two main groups—surface and volume-sensitive. In this review, we will present the advantage of the thickness shear mode (TSM) acoustic method and its variations for studying the activity of plasmin and trypsin at surfaces of piezoelectric crystals modified by short peptides or caseins. We already demonstrated the high sensitivity of acoustic sensors for the detection of the cleavage of β-caseins by trypsin and plasmin. The sensitivity of the TSM method is compared with those based on high-resolution ultrasonic spectroscopy (HR-US) that are convenient for studying the kinetics of the cleavage of caseins and other milk proteins by proteases in a small volume of approx. 1 mL.

The study of proteases activity is important also for obtaining information about the mechanisms of the enzymatic processes. Detection of the concentration of plasmin and its activity in milk is important for the dairy industry and, especially, for cheese makers. Among various methods of detection proteases, those based on acoustics principles are of a high advantage, because they do not need any labeling of the substrate. In this review, we will focus on the explanation of the principles of surface-based acoustics methods and their use for the detection of plasmin and trypsin. We will also explain the principles of detection of proteases activity in a volume using high-resolution ultrasonic spectroscopy (HR-US).

## 2. Milk Proteases

Bovine milk in its natural state is a white substance containing various components, mainly water (86%), lactose (4.7%), lipids (4%), proteins (3.2%), citric acid and minerals [9]. Its composition is dependent on the species and breed of individuals. The absence of certain protease systems causes changes in the amount of milk components and even can result in their absence. This can lead to the unique properties of milk in different mammal species [10]. Even some pathological states can support changes in milk due to effects on milk-producing cells. These circumstances are usually connected with inflammation processes [11].

Milk proteins are the most interesting and the most studied components of milk. They take part in the various processes in milk, as their functionality varies. These proteins are surface-active, which allows them to form self-assembled structures at various interfaces and micelles in a volume. Milk proteins are divided into several groups. Mass spectroscopy analysis proved that caseins are the most common proteins in milk. Caseins represent 78.3% of the total milk protein mass [9]. Serum proteins, often called whey proteins, take approximately 19%. The rest of the milk mass is presented by other protein groups, namely by membrane proteins and enzymes. Caseins are divided into several types, such as α_s1_, α_s2_, β and κ. Each of them has a different structure, functionality and location on the pseudo-micelles. Similar as with the protein group composition, the amount of each casein type differs in each species and breed of mammal. While in bovine milk, (*Bos Taurus*) β-casein is only the second-most common casein, in sheep milk (*Ovis aries*), the amount of β-casein exceeds other casein types. This contributes to the bitterness of sheep cheese, which was already affected by proteolysis [12].

Proteolytic systems are inseparable part of all kinds of milk. They regulate the protein content in milk, and they are also regulated. Like other proteins, they differ in all mammal species. It is not unique for certain protease systems to be absent in one species and to be the most prominent for another. Proteases of these systems are excreted as inactive zymogens. They are later activated by the cleavage of specific peptide bonds. There are several ways how milk can obtain proteases (Figure 1). Proteases can either get into the milk via the blood stream or are released by the mammary epithelial cells. Lastly, milk contains immune cells that can also release proteases into the milk environment [13].

Milk proteases are responsible for formatting the flavor of milk and milk products. This is mainly due to the degradation of casein molecules. Casein is prominently found in milk in the form of casein pseudo-micelles. These pseudo-micelles have complex structures. Each casein subtype has its own role and function. While κ-casein is usually found in the outer part of pseudo-micelles and protects it against degradation, β-casein is mostly located in the inner core [14] (Figure 2). If the pseudo-micelle is destabilized, proteases can cleave the unprotected β-casein. Cleavage can create several remnant peptides, called γ-casein fragments. These fragments tend to aggregate, making the texture of milk bulkier. Fragments also influence the flavor of milk, making it much bitter [15].

In general, the plasmin system is the most effective in the cleavage of milk proteins. It cleaves mainly α-casein and β-casein present in bovine and human milks. Plasminogen, the plasmin zymogen, is activated by the transformation to double-stranded active enzymes. This is achieved by the cleavage of the peptide bond in the plasminogen structure between Arg561 and Val562 [16]. Plasmin itself promotes plasminogen activation, causing a chain reaction. Even a small quantity of plasmin can support a prominent increase in cleavage [17]. However, the original activation is dependent on two activators: the urokinase plasminogen activator and tissue plasminogen activator. To prevent autonomic plasminogen activation by these proteases, there are several molecules influencing plasminogen activators. The presence of fibrin is important for the proper functionality of the tissue-type plasminogen activators, such as urokinase. The functioning of this activator is also affected by α_s2_-casein and κ-casein [18]. They bind to plasminogen, changing its conformation and making it more available for the activator to be cleaved and activated [19]. Increase of the activity of the plasminogen activator was observed in the presence of whey proteins, but the mechanism of this effect is still not clear [20].

Other regulation mechanisms of plasmin are plasmin inhibitors and the inhibitors of plasminogen activators. The most prominent is α_2_-antiplasmin. It is the most specific plasmin inhibitor. Milk also includes several nonspecific inhibitors, such as α_2_-macroglobulin [21]. α_2_-antiplasmin can create complexes with the plasmin, which even after separation, lead to a decrease of protease activity and inhibitor binding [17]. Plasminogen activator inhibitors belong to another specific regulatory system. Only the plasminogen activator inhibitors I and II were confirmed in milk [21]. UHT processing of milk must be therefore performed correctly (see [22] for the methods of UHT processing of milk). The inadequate heating of milk during UHT treatment, especially omitting the process of heating at 140 °C for at least 15 s, causes damage to the thermally unstable inhibitors of the plasmin system in milk [23,24]. Plasminogen activators in bovine milk are thermally more stable in the range of temperatures from 60 °C to 140 °C [25]. The increased activity of plasmin activators was observed after the thermal treatment of milk with a high content of somatic cells, but this effect was not completely monitored in the dairy industrial conditions [26]. The nonspecific inhibition of plasmin includes another group of milk proteins, whey proteins, namely α-lactalbumin, β-lactoglobulin and bovine serum albumin (BSA). The inhibition process of these proteins was not yet completely documented, as their inhibition effect is strongly dependent on the specific substrate that is the target of plasmin-induced proteolysis [27].

Plasmin is not an exclusive protease in milk. Another protease active in human milk is trypsin [28]. It is, however, not active in bovine milk. Trypsin concentrations in human milk are in the range of 2.9–5.6 μg/L, and its active form is present as an anion (trypsin-2). The activation of a trypsin anion is not completely clear, because for its zymogen, trypsinogen, the ways of its activation in milk are not known. Both trypsin activators, thrombin and enterokinase, are either absent or are found only in the form of zymogens. Therefore, trypsinogen is probably activated when the ingested milk reaches the duodenum. In the duodenum, it is cleaved and activated by a corresponding protein kinase. Trypsin activity is further inhibited by several inhibitors, namely α_1_-antitrypsin. The presence of α_1_-antitrypsin is lowered at the beginning and during lactation. It is active only in the small intestines of newborns. However, trypsin is there separated from α_1_-antitrypsin and other inhibitors [29].

The cathepsin protease system contains several cathepsins, and each of them differs in their function and presence of certain mammal species [30]. One of the cathepsin present in bovine and human milk is cathepsin D. Even when discovered in the active from, most of cathepsin D is in the form of an inactive zymogen. Cathepsin B, similarly to other proteases, cleaves mainly α_s1_-casein and β-casein. It, however, cleaves them at different peptide bonds. It is inactivated in basic pH, and its general purpose and activity in milk is unclear.

The kallikrein system has a prominent role for coagulation and fibrinolysis. Its active form is absent in milk, but some kallikrein molecules were detected in milk by proteomics [31]. The Western blot method was used to positively detect the plasma protease C1 inhibitor, which is important for blood clot formation.

In addition to the above-mentioned proteases, milk contains several other protease systems [13]. The elastase protease system was discovered in the neutrocytes of bovine milk. However, elastase activators are missing in milk, but several inhibitors were found. The chymotrypsin system depends on the chymotrypsin activator—trypsin. It is, however, absent in bovine milk. Colostrum contains several peptide sequences with a thrombin structure, but it lacks activators to release them. The aminopeptidase and carboxypeptidase protease systems were discovered by proteomic methods in mammary glands, but their important activators, including thrombin, are missing in this environment. The last identified protease system was matrix metalloproteinase. It degrades the extracellular matrix, allowing reconstruction of the tissue. Collagen and gelatin-degrading metalloproteinases MMP-2 and MMP-9 were localized in human milk, and the inhibitors for MMP-4 and MMP-1 were also discovered.

## 3. Surface Acoustic Methods

Piezoelectricity-based sensors are among most used sensing devices on the microscale and macroscale levels, as is evident from an increasing number of scientific papers that further broaden their applications [32,33]. Piezoelectric devices are generally divided into two groups: surface acoustic waves (SAW) and bulk acoustic waves (BAW). SAW contain electrodes at one side of the crystal. Generated wave deformation is defined by the crystal’s surface. While more sensitive, signals generated by devices suffer significant attenuation in biological solutions. More suitable biosensors for measurements in liquids are BAW-based sensors, namely quartz crystal microbalance (QCM) and TSM [34].

### 3.1. The Principles of QCM and TSM

The first biosensor was reported in 1962, when Leland C. Clark and Champ Lyons suggested the concept of the oxygen electrode with the immobilized glucose oxidase at the electrode surface for the detection of glucose [35]. The IUPAC (International Union for Pure and Applied Chemistry) defines biosensors as an integrated receptor-transducer device that can provide selective quantitative or semi-quantitative analytical information using a biological recognition element [36].

The first biosensors used were mostly electrochemical transducers. Other transducers, such as optical, mass or thermal reaction-based, are also currently and widely used. Mass-sensitive acoustic (piezoelectric) biosensors are based on the measurements of the properties of the acoustic waves that travels through the volumes and surfaces of sensors. The changes in the trajectory that waves travel causes changes in its velocity, amplitude and phase. Using these changes in gas or liquid environments, it is possible to measure changes of masses on surfaces caused by adsorbing or desorbing molecules. It is also affected by viscoelastic properties of thin films and liquid on the transducer surface. Acoustic sensors are rather popular due to their high sensitivity to detect small changes of masses at the surface. The resonant frequency of the crystal is dependent on its thickness, density and mechanical properties [37].

In 1880, Jacques and Pierre Curie demonstrated piezoelectricity. They showed that mechanical stress applied to certain materials (crystals, ceramics and bone) generated an electric difference potential [38]. This direct piezoelectric effect is used even in the recent development of novel Heel Strike generators, converting the mechanical energy of walking into electrical energy [39]. In contrast, deformation caused by the application of an electric field is called the inverse piezoelectric effect. In 1959, Günter Sauerbrey proved that changes of a mass on the quartz crystal surface are related to the changes of the oscillation frequency of quartz [40]. This resulted in his discovery of the quartz crystal microbalance (QCM) method. Sauerbrey’s work established the basis for the development of quartz oscillators and a sensitive microbalance for the measurement of thin film masses. Mechanical strain on the crystal surface induces the oscillation of certain frequencies in the crystal. This is electrically represented by the Butterworth-Van Dyke equivalent circuit (Figure 3A). In this circuit, the C_0_ is the static electrical capacitance. Due to the piezoelectric properties of the quartz, the electromechanical coupling resulted in additional motional elements L, C and R_m_ [41]. The parameters of this circuit can be determined by a network analyzer, as is schematically presented in Figure 4. Changes of the oscillation frequency, *Δf*, of the crystal are related to the changes of the mass, *Δm*, on the crystal surface, which is expressed by the Sauerbrey equation:*Δf* = −2*f_o_*^2^*Δm*/*A*(*μ_q_ρ_q_*)^1/2^(1)
where *f_o_* is the fundamental resonant frequency (Hz), *A* is the active crystal area (usually 0.2 cm^2^), *ρ_q_* is the quartz density (2.648 g.cm^−3^), *Δm* is the mass change (g) and *ρ_q_* is the shear modulus of the crystal (2.947 × 10^11^ g.cm^−1^ s^−2^).

Changes of the resonance frequency determine the mass of the material absorbed on the surface, usually in ng/cm^2^. The Sauerbrey equation is valid for elastic materials, such as metal coverings, metal oxides and adsorbed layers in vacuum. In this case, no loses in the energy during the oscillation occurred. It is, however, difficult to apply the Sauerbrey equation on the crystals covered by viscoelastic materials such as cells, polymers and complex biomolecular systems. The viscosity causes a loss of the oscillation energy, which affects the frequency changes. The viscosity contribution can be evaluated by determination of the motional resistance, *R_m_.* If the adsorbed mass on the crystal surface is heavier than 2% of the crystal mass, the Sauerbrey equation becomes invalid, and the linearity between *Δm* and *Δf* is lost [42].

Due to the above-mentioned problems in the determination of the adsorbed mass for the viscoelastic layers at the quartz crystal in liquid surroundings, a more precise term for QCM was established—the thickness shear mode (TSM). It defines the mode of the oscillations. For the development of the acoustic biosensor, a thin quartz crystal of a circular shape is used, with sputtered gold electrodes at both sides. Transducer geometry is a key factor [33]. The quartz disc must be cut under a specific angle in respect to the crystal lattice. AT or BT cuts are used to secure the perpendicular propagation of the wave. The AT cut is the most used and requires a 35°15′ cut towards the z-axis of the crystal. The influence of the temperature on the resonance frequency for the AT-cut crystal is almost negligible at 25 °C. The temperature coefficient causes minimal changes of the frequency at this temperature in the case of small temperature fluctuations. BT cut quartz crystal is approx. 50% thicker in comparison with AT cut [43].

Typically, the increase in the mass at the TSM transducer is accompanied by a decrease of the resonant frequency, as well as by changes of the motional resistance. The mass sensitivity of the crystal depends on its thickness. Thinner crystals have a higher resonance frequency and sensitivity. However, they are more difficult in manipulation due to their fragility. The relation between the resonant frequency, *f*_0_, and the thickness, *h*, of the crystal is given by equation:*f_o_* = *u*/2*h*(2)
where *u* = (*μ_q_ρ_q_*)^1/2^ is the sound velocity [44]. Typically, the crystals with a fundamental frequency in the range 5–10 MHz are used, which corresponds to their thickness in the range of 0.334–0.167 mm, respectively. Optically polished crystals are more suitable for work in the liquid phase due to less damping of the oscillations. Changes in the frequency and signal attenuations caused by the viscosity contribution make measurements more complicated and require advanced evaluation methods to separate these effects [45]. The TSM usually works in the flow setup. Changes of the main characteristics, the resonance frequency and motional resistance, can be monitored in real time by PC-controlled measurements (Figure 4).

The TSM is known by several modifications. One of the most prominent is quartz crystal microbalance with dissipation (QCM-D). Except the traditional frequency and resonance measurements, QCM-D allows measurements of the energy dissipation. The reason is that, thanks to the viscous forces, the acoustic wave that is propagated from the crystal into the surrounding solution decays (Figure 3B). The decay length, δ, can be determined by the following equation:*δ* = (2*η_L_*/*ωρ_L_*)^1/2^(3)
where *η_L_* is the viscosity, *ρ_L_* is the density of the surrounding liquid and *ω* = 2*πf* is the circular frequency of the oscillations. For crystal of a fundamental frequency of 5 MHz, the value of *δ* in water is 0.25 μm [44]. Only the processes at the surface of the quartz crystal that proceeds in the framework of the decay length can affect the series resonant frequency and motional resistance. In the QCM-D method, the crystals do not oscillate continuously. Oscillation of the frequencies close to the crystal resonance is induced with breaks. The oscillation generator produces quick jumps between several higher harmonic frequencies (see Jonsson et al. [46] for the principles of the QCM-D method). The level of the frequency response *Δf* is increased with the increasing of the harmonic number, *n:**Δf* = −2*nf_o_*^2^*Δm*/*A*(*μ_q_ρ_q_*)^1/2^(4)

The possible viscosity contribution was mathematically modeled and verified using the experimental data of the protein adlayers of the bilirubin oxidase [47]. It has been shown that, for conditions where the Sauerbrey equation is not valid, the algorithm developed help to find the responses that are best for detecting the viscoelastic changes of the adlayers. The algorithm also allowed increasing the time resolution of the experiments by recording the most important harmonics.

QCM, TSM and QCM-D-based biosensors were developed for a large number of assays connected with analysis of the molecular interactions at surfaces (see [48] for a recent review), as well as for the study of cell–surface interactions [49]. New trends consist in the application of DNA aptamers immobilized at the surface of TSM transducers for cancer diagnosis by the detection of cancer cells [50]. QCM sensors are suitable also for the detection of viruses [51].

### 3.2. The Principles of Electromagnetic Piezoelectric Sensors EMPAS

Another novel acoustic technique is electromagnetic piezoelectric sensors (EMPAS). This method combines the thickness shear mode (TSM) and magnetic acoustic resonance sensor (MARS) [52]. In this method, an extremely thin (approximately 83 μm) AT-cut quartz crystal with a 13-mm-diameter is placed in tight contact with the planar electromagnetic copper coil. The spacing does not exceed 30 μm. The current flowing in the coil produces an electromagnetic field, imitating the effect of the TSM. It generates a secondary electric field and, thus, mechanic oscillation on the piezoelectric material (Figure 5). The resonant frequency changes caused by the adsorption or desorption of the mass on the crystal surface induce the change in the coil impedance. This effect allows the monitoring of the kinetics of the frequency changes. The measured frequency is selected from one of the discrete odd multiples of the fundamental frequency, around f = 20 MHz. The EMPAS working frequency is typically 0.94 GHz, representing a 47-harmonic frequency. It is possible to even reach a working frequency of 1.06 GHz, corresponding to a 53-harmonic frequency [53]. The electromagnetic excitation mechanism produces an oscillation without gold electrodes at the crystal surface. The sensitivity of the EMPAS is high and allows monitoring of the surface processes, even in the case of the low concentration of the compound of the interest.

The EMPAS has demonstrated wide range of applications since its invention. Firstly, the adsorption of neutravidin layers was analyzed to compare the TSM and EMPAS [52]. Later, more complex tasks were studied. In particular, DNA aptamers MN4 selective to the cocaine that were immobilized at the surface of the crystal by crosslinking with S-(11-trichlorosilyl-undecanyl)-benzenethiosulfonate. The detection of cocaine with immobilized aptamers by the EMPAS was possible with a limit of detection (LOD) of 0.9 μM [54]. Using the EMPAS and the crystal with immobilized antibodies against HIV, the label-free detection of HIV-1 and HIV-2 was performed. The results showed specificity of the biosensor for anti-HIV-2 antibodies with a LOD of 100 μg/mL [55]. Similarly, anti-PTHrP protein antibodies were tested using the EMPAS. The PTHrP protein is potential biomarker for breast and prostate cancer. The immobilization of aptamers and testing of the protein allowed to achieve a LOD of 61 ng/mL of the PTHrP protein [56]. The EMPAS has been also used for the analysis of the interaction of milk and a serum with the surfaces of the crystals covered by antifouling layer. It has been shown that the layer composed of monolayer-forming surface linker 3-(3-(trichlorosilylpropyloxy) propanoyl chloride (MEG-Cl) provides excellent antifouling properties [57].

### 3.3. Immobilization of the Proteins at the Piezoelectric Transducers

For the detection of proteases activity at surfaces, it is crucial to optimize the methods of preparation of short peptide or protein layers that serve as a substrate for proteases of interest. The preparation of the protein layers on the surface of the transducers is among common applications of acoustic biosensors. For example, the preparation of casein layers is attractive for the study of their physicochemical properties and for applications in the pharmaceutical and food industries [58].

Interactions between the amyloid-β protein and different forms of vitamin D on the QCM surface were studied by Matsunage at al. [59]. They showed different effects of vitamin D2 and D3 on the oligomerization of amyloid-β proteins. The obtained results correlated with an electron microscopy study.

The protein layers were used also for the immobilization of biotinylated DNA aptamers in the development of acoustic aptasensors. In this case, the neutravidin layers are immobilized at the gold surface of the QCM transducer by chemisorption. This was accompanied by a strong decrease of the resonant frequency and rather small changes in the motional resistance, which is evidence of the formation of a rigid protein layer. In this case, the evaluation of mass changes and subsequent determination of the neutravidin surface density was made possible by the Sauerbrey equation. Changes in the frequency *Δf* = −196 Hz caused by the immobilization of neutravidin (Mw = 66 kDa) yield to a surface density of 1.2 × 10^12^ molecules/cm^2^ [60].

Protein immobilization was also used in the development of an EMPAS-based biosensor for the detection of the hexa-histidine-tagged heat shock protein 10, which is a biomarker for the early stage diagnosis of ovarian cancer. This protein was immobilized on the quartz discs surface by crosslinking with trichlorosilane. It was shown that this protein interacts specifically with DNA aptamers immobilized at the surface of the quartz disc [61].

Milk proteins were also immobilized at the surfaces of acoustic transducers. A hydrophilic silicon dioxide surface was used to immobilize several casein types. The adsorption of α-casein, β-casein and κ-casein with concentrations below the critical micellar concentration (CMC) was monitored by multifrequency QCM-D measurements. It was shown that β-casein and κ-casein were able to produce stabile layers. This is demonstrated in Figure 6, where the changes of the resonant frequency following additions of the caseins on the SiO_2_ surface with subsequent washing by a buffer are shown.

The fundamental and several higher harmonic frequencies were later used for a machine-learning algorithm to distinguish the adsorption of different casein types more effectively. The β-casein layer was tested also for the pH and salt stability. The pH stability tests suggest that β-casein is stable in lower pH (5–7.4). Alkaline pH caused a significant desorption of β-casein, resulting in a ~10-Hz frequency shift corresponding to a loss of ~25% of the film mass at pH 8 and a ~15-Hz frequency shift corresponding to a loss of ~35% of the film mass at pH 9. This information is important, as it could imply that β-casein is suitable for drug delivery in acidic environments [62]. Except for the casein layers, the short peptide monolayers were also used as a substrate for plasmin detection. The peptides with a primary structure: Lys-Thr-Phe-Lys-Gly-Gly-Gly-Gly-Gly-Gly-Cys that are cleaved by plasmin at the Lys residue close to the Gly, are immobilized at the QCM transducer by chemisorption. This was possible due to the Cys residue at the C end of the peptide. The applications of these layers for plasmin detection are discussed in Section 3.4. [63].

The EMPAS was also used for monitoring the adsorption of β-casein with concentrations below the CMC (<0.5 mg/mL) on hydrophilic and hydrophobic surfaces. It was shown that β-caseins were better adsorbed on hydrophobic surfaces in comparison with hydrophilic surfaces. In addition, the hydrophobic surfaces were much more stable. The orientation of β-casein molecules at hydrophilic surfaces caused the aggregation of air microbubbles at the surface, which is not desirable in acoustic measurements [64].

### 3.4. Application of Surface Acoustic Methods for Detection Proteases

The surface of the acoustic biosensor modified with a suitable substrate can be used for the detection of protease proteolytic cleavage. One of the first methods that used QCM for protease detection was the work by Sabot and Krause [65]. They used thin polymer films as the substrate for the detection of chymotrypsin and dextranase. This paper showed the differences in the mechanisms of degradation caused by proteases and other factors, namely pH. Human neutrophil elastase (HNE) and cathepsin detection were the aim of the paper by Stair et al. [66]. High levels of HNE are associated with the inflammatory states of several acute and chronic diseases. They used peptide crosslinked dextran hydrogels as substrates immobilized on the surface. They demonstrated a direct relationship between the hydrogel degradation rate and HNE activities from 2.5 to 30 U/mL. Film degradation was rapid and complete in less than 10 min for HNE activities ≥ 10 U/mL. With increasing the crosslinking density from 25% to 75%, the rate of degradation increased by 3.5-fold. Another peptide crosslinked poly(ethylene glycol) hydrogel substrate was used for the detection of collagenase by means of QCM. This is important for the diagnosis of pathological diseases such as rheumatoid arthritis and osteoarthritis. The sensor allowed the detection of collagenase in the range from 2 nM to 2 μM [67]. Huenerbein et al. [68] used β-casein layers covalently immobilized at the gold electrodes of piezocrystals for the detection of trypsin and pepsin. They were able to monitor the cleavage of β-casein by both proteases in real time. The cleavage was pH-dependent and accompanied by an increase of the resonant frequency of the piezoelectric crystals. The LOD was, however, not reported in this paper.

Acoustic methods were also used for the detection of bacterial proteases. The ClpYP protease can be found in *Escherichia coli*. Bacterial protease-induced protein regulation can be related to antibiotic resistance or to other important factors. The QCM method revealed a strong interaction between ClpYP and YbaB proteins [69].

The TSM method was also used for the detection of milk proteases. The first research was focused on the deposition and cleavage of specific peptide substrates chemisorbed at the piezoelectric crystal. The method allowed the detection of plasmin with a LOD of 0.65 nM (Figure 7) [63].

Considering that the concentration of plasmin in raw milk is in the range of 4–12 nM, this method is suitable for practical applications. The multi-harmonic QCM has was for the detection of trypsin and plasmin by monitoring frequency changes following the cleavage of β-casein layers immobilized at the SiO_2_ surface. This method allowed the detection of the proteases with high sensitivity: 0.2-nM trypsin and 0.5-nM plasmin. This paper also demonstrated an advantage of application of machine-learning algorithms to distinguish between trypsin and plasmin [62]. Similar layers were later used for the detection of plasmin by the EMPAS. Hydrophilic and hydrophobic β-casein layers were immobilized on the crystal discs, and different concentrations of plasmin were applied. As hydrophobic β-casein produced more β-casein on the crystal surface, a much stronger kinetic response of frequency changes during plasmin applications were observed. This method allowed to achieve a LOD of plasmin detection at a very low concentration: 32 pM [64].

## 4. High-Resolution Ultrasonic Spectroscopy (HR-US)

The direct real-time monitoring of biochemical processes at their natural state and native media, as well as in a wide range of processing conditions, is an important tool in research analytical laboratory work. There are a wide range of sensors available that provide direct insight into the bioprocess states in terms of physical, chemical and biological variables. They are also preferred for the efficient process control of the manufacturing of final products and process optimization. Traditionally, the real-time detection of bioprocesses in the volume phase has been the convenient approach for most of the routine analytical applications due to its feasibility. It still presents a powerful detection modality in most homogenous activity assays to characterize the protease action, as well as properties of the peptide products [70]. In addition, detection in the volume phase also diminishes the need for extended sample preparation, which may possibly lead to alterations of the structures of the proteins. It has been desirable to have a sensitive and selective analytical technique that is capable of detection and monitoring in the volume phase. Traditional methods employ optical methods due to remarkably high sensitivity towards optically active atomic groups. Optically inactive samples such as proteins require conjugations or derivatization with optically active marker. However, this often results to alteration in properties of the molecules. They are also limited sometimes with the stability of the molecular markers and structural complexity of the media.

Alternatively, within the last decades, low-intensity ultrasonic techniques, particularly ultrasonic velocimetry and spectroscopy, have been classified as nondestructive mechanical sensors for the online characterization of the state of biomolecules, as well as biochemical reactions, in native or processing media [71,72]. These techniques offer sensitivity measurements of the viscoelastic properties of analyzed samples determined by the molecular forces and dynamics involved in the process. Since most of the biochemical samples are ultrasonically transparent, ultrasonic measurements can be performed in a wider range of solutions without the need for a chemical probe, as well as in opaque and high-concentration samples that are difficult to measure with optical methods.

Among the recent ultrasonic techniques, high-resolution ultrasonic spectroscopy (HR-US) displays and offers numerous advantages over the optical methods and traditional ultrasonic technology, particularly for the real-time noninvasive precise monitoring of proteolysis reactions towards milk proteins by milk proteases in a volume. The capabilities of HR-US in monitoring a wide range of biocatalysts have been comprehensively described in reference [73]. However, examples of the HR-US application in a milk protein-protease system are very few. The HR-US monitoring of bioprocesses is based on the simultaneous measurements of changes of the characteristics of two major ultrasonic wave parameters: ultrasonic velocity, u, and ultrasonic attenuation, α, during a wave propagation caused by a change in the molecular characteristic. These ultrasonic parameters have different sensitivities to the physicochemical characteristics of molecules and their processes. Recent ultrasonic methodology emphasizes the experimental methods and data algorithms on translating both ultrasonic parameters into more useful quantitative biochemical transformations, such as real-time profiles of the evolution of the number of peptide bonds hydrolyzed per unit of time by means of calibration methods [73]. These real-time ultrasonic profiles are useful to provide kinetic and mechanistic descriptions of the proteolysis of milk proteins in terms of kinetic and thermodynamic parameters consistent with the enzyme kinetic models, as well as for process control and optimization.

### 4.1. Principles of HR-US

Ultrasonic spectroscopy (US) is a continuous ultrasonic wave-based analytical technique that utilizes a low amplitude, longitudinal deformation of high-frequency (1–20 MHz) acoustic waves propagating through the analyzed sample [71]. In contrast to traditional optical spectroscopic techniques, US probes the viscoelastic, rather than electric and magnetic, characteristics of the analyzed samples. Thus, most liquids have ultrasonic transparency. Among the measuring methods of an ultrasound, it belongs to a class of resonant ultrasonic-type techniques consisting of a fixed couple of emitting–receiving piezoelectric transducers (LiNbO_3_) located at opposite side walls of a resonator chamber with a pathlength, d (Figure 8A) [74]. In the resonance chamber, the analyzed sample is typically filled in. The emitting transducer generates and propagates the ultrasonic wave through the sample and towards the receiving transducer. Following this, the propagating waves are reflected and forth through the sample multiple times, creating a positive interference. By principle, resonance occurs when d is an integer number, n, of the half wavelength, λ, i.e., $d=n\frac{\lambda }{2}$. The interference creates a higher amplitude resonance peak at frequency, $f_{n}$, which is detected and measured. This provides the high-precision measurements of ultrasonic wave characteristics such as the ultrasonic velocity, u
$\left(=f_{n}\frac{2d}{n}\right)$, determined from the resonance frequency of the solution, and ultrasonic attenuation, α, which is the ultrasonic energy losses given by the bandwidth of the resonance peak.

There are two kinds of low-frequency ultrasonic waves (in the MHz frequency range), longitudinal and shear wave, which can be generated by the vibrating surface of a piezoelectric transducer in contact with the analyzed medium [75]. Currently, a longitudinal wave has been solely utilized for the purpose of sensing biocatalytic reactions in a volume. In a longitudinal ultrasonic wave, the direction of the propagation of the ultrasonic wave along the x-axis is parallel to the direction of the oscillation of the molecules and their molecular arrangements. This results in an oscillation displacement, X(x,t), of the medium from position of the medium at unstrained state x and time t. The resulting acceleration of the medium and mechanical momentum give rise to the well-known wave Equation (5), describing the propagation of the longitudinal wave in an isotropic medium, which is homogenous within the scale of an ultrasonic wave of low amplitude:(5)$$\frac{\partial^{2}X(x,t)}{\partial t^{2}}=\frac{M}{\rho_{I}}\frac{\partial^{2}X(x,t)}{\partial x^{2}}$$
where M is the modulus of longitudinal deformation, and $\rho_{I}$ is the effective inertial density of the medium. The detailed mathematical formulation and physical description of the propagation of a longitudinal ultrasonic wave have been previously well-discussed in the references [71,76,77,78]. Furthermore, under oscillation conditions where there are time delays between the stress and the strain, the displacement X possess a complex characteristic [79]. Solving the differential equation above yields the following complex solution: $X=X_{0}e^{-\alpha x+i\omega \left(\frac{x}{u}-t\right)}$, where $X_{0}$ is the amplitude of the wave at x = 0 and t = 0, ω(≡2πf) is the angular frequency and $i\left(\equiv \sqrt{-1}\right)$ is the imaginary unit. Consequently, M = $M^{\prime }+iM^{\prime \prime }$. In liquid and gel systems, the elastic nature is dominant; thus, the imaginary loss modulus, $iM^{\prime \prime }$, is usually very small: $iM^{\prime \prime }<<M^{\prime }$. Following this, from the mathematical relationships discussed above, the two determinants of the propagation of longitudinal waves through liquid mixtures are ultrasonic velocity and attenuation and are given as [78,80]:(6)$$u=\sqrt{\frac{M^{\prime }}{\rho_{I}}};\;\alpha =\frac{\omega M^{\prime \prime }}{2\rho_{I}u^{2}}$$

The ultrasonic parameters are related to the molecular characteristics governing the viscoelastic properties of the materials by the following: $\frac{M}{\rho_{I}}=\frac{u^{2}}{{\left(1-i\frac{\alpha \lambda }{2\pi }\right)}^{2}}$. The modulus M $\left(\equiv K_{V}+\frac{4}{3}G\right)$ is the superposition of the volume deformation, $K_{V}$, and the modulus of the shear deformation, G. The term volume deformation $K_{V}$ is more convenient to express in terms of the coefficient of adiabatic compressibility β($\equiv \frac{1}{K_{V}}=-\frac{1}{V}\frac{\delta V}{\delta P}$), which represents a relative change of the volume, δV, of the system per unit of pressure applied, δP. For a simple homogenous protein solution, e.g., in the buffer system, the parameter G is significantly smaller than $K_{V}$ and can be neglected. As a result, the ultrasonic velocity in Equation (6) can be simplified into $u=\frac{1}{\sqrt{\beta \rho_{I}}}$. Since the radius size of the protein molecules and particles, 1–20 nm, are normally smaller than the wavelength of the viscoelastic waves at an ultrasonic frequency range, e.g., λ~100 µm at 15 MHz, the $\rho_{I}$ shall be equal to the density of the protein solution, ρ(=$\frac{1}{v}$), where v is the specific volume and $\beta_{S}$ = $\rho k_{S}$, where $k_{S}$ is the specific adiabatic compressibility, and the subscript “S” refers to the entropy. This results in a known mathematical relationship for ultrasonic velocity in a homogenous protein system, $u=\frac{v}{\sqrt{k_{S}}}$.

The major determinant of the ultrasonic velocity is the compressibility (viscoelasticity) property of the medium. Compressibility is extremely sensitive to the intermolecular interactions, as well as the molecular organization taking place during a biomolecular process. During the oscillation of the wave, there are fluctuations of the pressure and temperature that result in the compression and decompression dynamics of molecules. An example of this, which is crucial for the detection of biocatalytic processes, is the movement of water molecules within the hydration layer surrounding the polar atomic groups (e.g., amino group in Figure 8B) during the ultrasonic oscillation. The intermolecular forces attributed to the degree of resistance in the associated volume change as a response to the temperature-pressure change in the ultrasonic wave. The pressure-induced volume change produces a thermal fluctuation in the solute and continuous medium. Due to the difference in the physical properties of both systems, the amplitude of the thermal oscillation shall be different in between. Consequently, the heat exchange is generated from the effective boundary of the solute into the continuous medium. However, in the HR-US measurement, the periodic ultrasonic compressions are usually faster than the speed of the heat dissipation from the compressed volume, and the heat exchange in the interface is almost absent. In this case, the adiabatic condition in all ultrasonic measurements shall be presumed. In addition, the low-amplitude oscillation of the temperature and pressure provides the noninvasive characteristics of the measurement.

For complex media having viscoelastic characteristics, the propagation of the longitudinal wave is also determined by the shear storage modulus (measurement of G = $G^{\prime }+iG^{\prime \prime }$ in a complex media is possible). The loss modulus, $G^{\prime \prime }$, becomes significantly higher than $G^{\prime }$ at higher frequency (ultrasonic range) in contrast to lower frequency (dynamic rheology range). It is attributed to energy dissipation (viscous and thermal losses in these waves) in the form of scattered waves. In ultrasonic measurements, the energy losses in compressions and decompressions is represented as ultrasonic attenuation [81,82]. In a homogenous protein solution, ultrasonic attenuation is usually caused by heat dissipation to the medium dependent to its intrinsic properties (density and viscosity). In some cases, the ultrasonic wave is attenuated through the perturbation of fast chemical kinetic reactions and gives rise to frequency-dependent relaxation contributions. In a more complex nonhomogeneous sample, the presence of a dispersed particle with a significant radius size gives rise to ultrasonic scattering. The information on the ultrasonic scattering phenomenon can be utilized in particle size characterization.

### 4.2. Key Advancement and Attributes of HR-US

The recent advances in ultrasonic spectroscopy for monitoring biocatalysts utilized today are based on the advanced principles of ultrasonic resonator techniques developed a couple of decades ago, as well as the optimal built-in resonator cell designs. The current resonator instrumentation is capable of exceptionally high-resolution ultrasonic velocity, down to 0.2 mm/sec, and an ultrasonic attenuation measurement of 0.2% [71,73]. This allows the resolving details of small changes in the molecular characteristics in the multi-steps or side reactions of bioprocesses, e.g., ionization reactions between titratable groups in protein solutions and the occurrence of protein aggregation concomitant with hydrolysis. The lower sample volumes that require ranging from 0.03–2 mL offer an advantage, particularly in the use of expensive biological or chemical reagents. In addition, its spectroscopic classification comes from the generation of an ultrasonic resonance spectrum and the analysis of resonance peaks, within the range of 1–20 MHz, of the analyzed samples, as well as the transducer. Such broad frequency range measurements give rise to the analysis of ultrasonic relaxation and scattering effects concomitant with biocatalysts.

The current HR-US resonance cell composites and compartment geometry were optimized for the use of end-users withstanding a broad range of temperatures of −40 °C to 150 °C and elevated pressure (up to 20 bars). The cells are designed for easy filling and the retrieval of a wide range of liquids and gel samples, as well as appropriate cleaning and drying procedures. A common protocol to activate the hydrolysis of a protein in a volume is the direct addition of enzyme stock solution through the injection hole of the cap fitted with a rubber septum and followed by proper homogenous stirring. In addition, the cell is compatible for the use of nonpolar solvents. Thus, the study of biocatalysis in system like water-in-oil microemulsion [83] can be done. High-sensitivity measurements may operate in a clean sample at the absence of impurities, which can be eliminated through sample filtration. Efficient degassing of a protein solution is part of the liquid sample preparation to minimize the water-air interfaces created by solubilized gases in the sample. Furthermore, a high thermally controlled cell is also very crucial for high sensitivity of the measurements. The presence of temperature fluctuations, particularly for the ultrasonic velocity, can lower the sensitivity of the measurements. For example, in water, the ultrasonic velocity varies approximately 3 m.s^−1^ K^−1^ [84]. In the current instrumentation, differential cell modality is often utilized to compensate the temperature noise, and it is thermally controlled by a fluid circulator providing a maximum temperature stability of ±0.005 °C. The additional functionality of HR-US; suitably for titration, static and flowthrough and kinetic and temperature ramp measurements come with analytical accessories, such as precision temperature controllers, titration accessories and pressure and vacuum pumps for the titration methods that can be equipped within. These accessories allow the different measurement modes of the chemical processes.

### 4.3. Kinetics of Protein Hydrolysis

The enzymatic hydrolysis of a protein is a complex process because of the high specificity of protease and the diversity of the substrates involved [85], accessibility of the peptide bonds [86] and sensitivity to hydrolysis conditions [87]. Regardless of the complex factors, the reaction pathway of the enzymatic hydrolysis of a protein may be simply presented as below [88]:(7)$$\begin{array}{c} E+S\begin{array}{c} \overset{k_{ES}}{\to }\\ \underset{k_{-ES}}{\leftarrow } \end{array}ES\overset{k_{h}}{\to }E+P^{\prime }+P^{\prime \prime }\\ \;+\\ \;E\\ \;{↓}^{k_{d}}\\ \;E+E_{d}+S \end{array}$$

Initially, the enzyme E will bind to protein substrate S to reversibly form an enzyme–substrate complex ES, where $k_{ES}$ and $k_{-ES}$ are the rate constant of the complex formation and dissociation, respectively. This is followed by the irreversible hydrolysis of S into two products, $P^{\prime }$ and $P^{\prime \prime }$. Under some conditions, either S, $P^{\prime }$ and $P^{\prime \prime }$ may act as a reversible or irreversible inhibitor to the enzyme, $E_{d}$. The parameter $k_{h}$ (min^−1^) and $k_{d}$ (kg mmol^−1^ min^−1^) are the rates of enzymatic hydrolysis and deactivation (assuming that the enzyme denaturation is a second-order reaction), respectively. For milk proteases (e.g., plasmin) that display a high degree of specificity, the specific/hydrolysable peptide bonds act as the true substrate rather than the intact protein [73,87,89]. Their proteolytic action is on the peptide bonds, resulting in the liberation of C-terminal and N-terminal groups of protein hydrolysates:(8)$$P_{1}-C(O)-NH-P_{2}+H_{2}O\overset{k_{P_{1}-C(O)-NH-P_{2}}}{\to }P_{1}-COO^{-}+NH_{3}^{+}-P_{2}$$
where $P_{1}-COO^{-}$ and $NH_{3}^{+}-P_{2}$ are the protein hydrolysates with the C-terminal group and protein hydrolysates with the N-terminal group, respectively, $P_{1}-C(O)-NH-P_{2}$ represents the peptide bond and $k_{P_{1}-C(O)-NH-P_{2}}$ is the reaction rate constant of a specific peptide bond. The reaction rate constant may vary between specific cleavage sites due to the degree of accessibility, as well as the energetics of peptide bonds [90].

The kinetic progress of the reaction of the hydrolysis of a protein can be represented by the evolution of a concentration of peptide bonds hydrolysed, $c_{bh}$, which are measured by traditional enzymatic assays employing optical spectroscopy, the pH stat technique, a chromatography method and osmometry [73,90]. These variables require information on the substrate consumption or product formation with the time of hydrolysis. Thus, real-time and precision experimental techniques or protease assays for both the detection and monitoring of the reaction are often required for such detailed characterizations of the proteolytic activity. The extent of the reaction of protein hydrolysis is commonly represented by the degree of hydrolysis. The degree of hydrolysis ($d_{h}$) of proteins is defined as the percentage of the number of peptide bonds cleaved over the total number of peptide bonds present in the protein. Traditional quantitative methods to determine the degree of hydrolysis of proteins were based on the optical determination of the chemically derivatized free amino groups released during hydrolysis or the titration of protons released during a peptide bond hydrolysis. Optical assays are highly sensitive and reliable methods but are often discontinuous and invasive. They require the chemical derivatization of a terminal amino group with an appropriate optically active reagent, such as trinitrobenzene sulfonic acid (TNBS), ninhydrin and o-phthalaldehyde (OPA). However, there are limitations on the structural complexity of the media and the multistep measuring procedures, which restricts their precision and rises the cost of analysis [73]. On the other hand, the pH stat technique is simpler and a more direct assay, as it is based on the principle of pH titration. The quantification of the number of terminal end groups liberated from the hydrolysis is calculated based on the amount of NaOH or HCl consumed into the reaction mixture to keep the pH constant during enzymatic hydrolysis [91,92,93,94]. The reliability of the pH stat technique is in a mathematical relationship between the consumption of the acid/base and the number of the terminal end groups being analyzed. However, such a relationship exhibits complexity due to the presence of weak acids in the mixtures, as well as determining the actual value of the apparent constant of proton dissociation, $pK_{A}^{app}$ [90]. Other analytical methods that provide other descriptors of the extent reactions include reverse-phase high-performance liquid chromatography (RP-HPLC) and Fourier-transform infrared spectroscopy (FTIR). Both methods have been utilized to concomitantly determine the degree of hydrolysis, protein conversion rate and molecular weight distribution and explore their complementary natures [94,95,96,97,98]. Both methods are invasive, time-consuming and tedious in sample preparations and data collection and interpretation. An alternative to these methods, the HR-US technique, is based on the precision measurements of the significant hydration change concomitant with the evolution of the peptide bonds hydrolyzed. Thus, it directly quantifies the end terminal groups released through the change of hydration relative to the parent amide bond. HR-US overcomes most of the major limitations of the current electromagnetic wave techniques, making it a very potential technology for proteolytic activity profiling under different reaction conditions, as well as the development of a protease assay using natural substrates in the native environment. In addition, the ultrasonic profiles of hydrolysis can be translated into more useful biochemical information on hydrolysis [73,99]. The verification, as well as calculation, algorithm of the ultrasonic methodology will be discussed in the following section.

The application of ultrasonic measurements for the analysis of enzyme kinetics requires extremely high resolution and the ability to work with small volume samples. The remarkably high precision profile of the concentration of the peptide bond is derived into ultrasonic reaction rates using the numerical differentiation method or function differentiation. The resulting overall reaction rate is fitted with the enzyme model, which provides the kinetic and mechanistic descriptions of hydrolysis. One of the most widely accepted enzyme kinetic models is the classical Michaelis-Menten kinetic, which was originally developed to describe the enzymatic conversion of a single substrate to a single product. The approach uses the substrate concentration dependence of the initial rate of hydrolysis to determine its two important kinetic model parameters: the Michaelis–Menten constant, $K_{M}$ ($\equiv \frac{k_{-ES}+k_{h}}{k_{ES}}$), and the maximum rate of peptide bond hydrolysis, $r_{\max}$. Up to now, many researchers still suggest that protein hydrolysis obeys the Michaelis-Menten behavior in which the model parameters provide useful kinetic characteristics [87,91,100,101]. The $K_{M}$ and $r_{\max}$ are linked to useful kinetic parameters, such as the catalytic constant (also known as the “turnover number”), $k_{cat}\left(\equiv \frac{r_{\max}}{\left[E\right]}\right)$, and is defined as the maximum amount of substrate converted to a product per enzyme molecule per second and the catalytic efficiency, $\frac{K_{M}}{k_{cat}}$, which describes how well the proteins are bound to the protease. It can be argued that the standard Michaelis-Menten model cannot be applied to proteolysis due to the presence of specific peptide bonds, and the model assumes that these peptide bonds are hydrolyzed with the same kinetic constants and can be freely attacked by proteases. Consequently, many variants of the Michaelis-Menten model have been formulated, satisfying the description of the complex protein hydrolysis system.

An alternative approach is to perform a regression analysis of the degree of hydrolysis profile at a single substrate concentration to describe both the initial and later stages of hydrolysis. One such empirical kinetic model was proposed by Gonzalez-Tello et al. [102,103] that utilizes the exponential function with two fitting parameters related to $k_{h}$ and $k_{d}$. The exponential model was derived from Equation (7), which highlights the empirical explanation of the deceleration of $d_{h}$ at the later stage of hydrolysis, as well as the reduction of the maximum $d_{h}$ with increasing the substrate concentration. These could be attributed to three possible factors: (i) a decrease in the concentration of hydrolysable peptide bonds, (ii) inhibition of the enzymes by peptide intermediates and (iii) the physicochemical inactivation of enzymes. Furthermore, a comparison of the exponential fitting of the degree of hydrolysis obtained at different hydrolytic comparisons can be used to examine comprehensively the enzyme performance. The use of the model with HR-US to characterize the hydrolysis of β-casein by trypsin was previously reported [104]. The degree of the hydrolysis profile obtained was consistent with the exponential model description discussed above. The trypsin activity was characterized by comparing the kinetic parameters at different pH, temperature and enzyme concentrations. The temperature dependence of trypsin activity was shown to be exponential, thus obeying the classic Arrhenius model, at a small window of temperatures: 15–45 °C. Following this, the Eyring transition state theory (TST) was applied to understand the temperature dependence of the reaction rate constants and relate them to the thermodynamic parameters of the activation of proteolysis, such as the changes in entropy, $\Delta S^{‡}$, enthalpy, $\Delta H^{‡}$, and Gibb’s free energy, $\Delta G^{‡}$. It is also important to note that the classic model is somehow limited to a small temperature window. The consideration of the other thermodynamic variables and more complex temperature-dependent factors, such as the presence of denaturation, as well as aggregations, deviates from the model [105,106].

In a more advanced approach, the detailed ultrasonic “reaction rate vs. concentration of the reactant/product” profiles can also be applied for the establishment or verification of kinetic models, as well as the determination of the underlying kinetic parameters (relative activity rates, enzyme inhibition constants, maximum rate of hydrolysis and the Michaelis-Menten constants) and thermodynamic constants (activation and deactivation energies of the enzyme reaction), including those involved in complex mechanisms of inhibition [101]. This was successfully applied in the hydrolysis of disaccharides such as cellobiose [107] and lactose [108]. Resa et al. [107] used ultrasonic reaction rate profiles to propose a mechanistic model explaining the complex kinetics of the hydrolysis of cellobiose by β-glucosidase. In the model, the parameters of the inhibition effects, as well as the mathematical term related to a specific mechanistic pathway, were included. However, in the protein system, they are multiple substrate arrangements represented by peptide bonds. So far, the models discussed above assume that all intermediate peptides have a similar enzyme affinity and all the peptide bonds are hydrolyzed with the same kinetic constants and can be freely attacked by proteases.

A two-state kinetic model was proposed to explain the variations of the hydrolysis constant of individual peptides influenced by enzyme selectivity and accessibility. The model assumes the consecutive demasking and hydrolysis stages. The demasking process leads to the exposure of more peptide bonds, and the polypeptide chains are extensively degraded to different intermediate peptides in accordance with the kinetic characteristics of the enzyme. The ratio between the rates of the first and second stages of hydrolysis points out two proteolytic mechanisms proposed by Linderstrom–Lang: “one-by-one” (the rate constant of demasking is lower than hydrolysis) and “zipper” (the rate constant is higher than hydrolysis) types. In the “one-by-one” mechanism, masking of the peptide bonds was pointed out to be the main limiting factor of the hydrolysis of proteins. Vorob’ev et al. [109] reported that the kinetics of the initial stage of the hydrolysis of milk proteins by α-chymotrypsin and trypsin followed the so-called “one-by-one” mechanism. The authors verified of demasking process of β-casein and β-lactoglobulin by trypsin by the monitoring of tryptophan fluorescence during proteolysis. Buckin and Altas [73] also confirmed the “one-by-one” mechanism of α-chymotrypsin towards the degradation of β-lactoglobulin using the correlation between the ultrasonic hydrolysis profile and concentration of the peptide bonds hydrolyzed. The observed linear dependence indicates that the polypeptides with intermediate degrees of hydrolysis are nearly absent in the reaction mixture. A similar observation was reported in the ultrasonic analysis of the hydrolysis of β-casein by trypsin [104]. The two-step model contained the relevant kinetic parameters, the second order hydrolysis constant and Michaelis-Menten constant, which both were a function of the degree of hydrolysis due to the contribution of enzyme deactivation and variance in the productive binding of enzymes at different the substrates present. Therefore, these parameters can be quantitatively determined by fitting the reaction rate vs. the degree of hydrolysis with the model equation [86]. However, information on the ratio of the rate constants of demasking and the hydrolysis of unmasked bonds and the initial degree of masked bonds are required. Such tasks are complex, as they may require either reference data or another set of kinetic experiments.

### 4.4. Detection of Proteases Activity by HR-US

The ultrasonic detection of protease activity is based on the changes of the compressibility and density of the mixture caused by the hydrolysis reaction, affecting the characteristic of the ultrasonic wave traversing the mixture [71,73,108]. The enzymatic hydrolysis of one peptide bond, Equation (8), is accompanied by a change in the intrinsic properties of the protein (loss of the native or aggregate structure) and the interactions between the protein and their hydrolysates with the environment, which includes solvation effects, i.e., hydration. The released atomic groups, $–NH_{3}^{+}$ and $–COO^{-}$, subsequently interact with bulk water molecules through charge dipole interactions, and these results in the formation of a more rigid molecular network of water molecules, i.e., a hydration shell, around these atomic polar groups [107]. Therefore, the hydration level of the released atomic group experiences shall be greater than the parent amide bonds. Since the hydration layer is less “compressible” than that of the bulk water, this should result in a negative value of the change in hydration compressibility. The level of influence exerted on water in the hydration shell, i.e., electrostriction, and, hence, the magnitude of this decrease in compressibility depends significantly on the nature and the number of exposed atomic groups [107,109]. The contribution from the change in the intrinsic volumes due to the loss of the native structures of the proteins is relatively small compared to the hydration effect. Therefore, the negative contribution of hydration compressibility shall lead to the overall decrease of compressibility of the hydrolysis mixture.

The evolution of the concentration of polar end terminal groups during hydrolysis results in a greater hydration effect in the hydrolysis mixture, which can be precisely measured by HR-US. The decrease of compressibility during the hydrolytic processes leads to an increase in ultrasonic velocity observed in Figure 9 in the trypsin hydrolysis of β-casein in 0.1M phosphate buffer, pH 7.8, performed at 25 °C [104]. Thus, the mathematical relationship between the change in ultrasonic velocity in the reaction mixture, δu(t), during time interval δt at reaction time t and the change in the concentration of bonds hydrolysed $\delta c_{bh}(t)$ is described below:(9)$$\delta c_{bh}(t)=\frac{\delta u(t)}{u_{0}\Delta a_{r}}$$
where $\Delta a_{r}$ is the change in the concentration increment of the ultrasonic velocity of the hydrolysis reaction, defined as:(10)$$\Delta a_{r}=\frac{1}{u_{0}}\frac{du}{dc_{bh}}$$
$u_{0}$ is the ultrasonic velocity in the reaction medium without reactants and products. In summary, the magnitude of the change of the ultrasonic velocity is mainly determined by the difference in the hydration characteristics and in the intrinsic volumes of the atomic groups of reactants and of the products affected by the reaction.

The change of concentration increment of the ultrasonic velocity, $\Delta a_{r}$, in Equation (10) acts as the proportionality coefficient between the ultrasonic velocity and the concentration of the peptide bonds hydrolyzed. In a chemical reaction undergoing transformation from reactants to products, it is expressed as the difference between the concentration increment of the products, $a_{P}$ ($\equiv \frac{u_{P}(t)-u^{0}}{u_{0}c_{P}}$), and of the reactant, $a_{r}$ ($\equiv \frac{u_{R}(t)-u^{0}}{u_{0}c_{R}}$), i.e., $\Delta a_{r}=a_{P}-a_{R}$ [73]. The physical definition of $\Delta a_{r}$ is related to the hydration characteristics and intrinsic properties of the atomic groups of the reactants and of the products affected by hydrolysis [107,108]. The detailed molecular interpretation of the concentration increments of ultrasonic velocity, $a_{i}$, of hydrolysis reaction at a wider range of concentrations is discussed in detail [78,107]. In the proteolysis in the buffer, where infinite dilution is assumed, the $\Delta a_{r}$ is expressed as in Equation (11):(11)$$\Delta a_{r}=\frac{1}{2}\left(2\frac{\Delta \phi V}{v_{0}}-\frac{\Delta \phi K_{S}}{k_{S0}}\right)$$
where ΔϕV ($\equiv \frac{v(t)-v^{0}}{c_{bh}(t)}$) and $\Delta \phi K_{S}$ ($\equiv \frac{k_{S}(t)-k_{S}{}^{0}}{c_{bh}(t)}$) are the apparent molar change volume and compressibility of the hydrolysis at the infinite dilution case, where v and $k_{S}$ are the partial specific volume and adiabatic compressibility of the hydrolysis mixture at time t and time zero (superscript “${}^{0}$”), respectively, and $v_{0}$ and $k_{S0}$ are the specific volume and compressibility of the bulk medium. The ΔϕV and $\Delta \phi K_{S}$ are both experimentally measured using a density meter and ultrasonic velocimeter.

Ultrasonic calibration methods that have been utilized to experimentally determine the value of $\Delta a_{r}$ were previously outlined and discussed in detail by Buckin and Altas [73]. Among the four calibration methods discussed by the authors, “Method 2” was solely applied in the system of protein hydrolysis. The “Method 2” calibration involves the parallel ultrasonic velocity measurement with a discontinuous orthogonal method that measures the degree of hydrolysis at each time point. Orthogonal methods that have been utilized in the calibration of ultrasonic profiles of protein hydrolysis include the conjugation of fluorescent dye, boron-dipyrromethene (BODIPY), with casein aggregates for fluorescence measurements [110], ninhydrin [110,111] and TNBS methods [73,104] for UV-Vis measurements of the hydrolysis of bovine casein by trypsin and β-lactoglobulin by α-chymotrypsin and β-casein by trypsin, respectively. The TNBS method is considered a milder and more reliable reference method to ultrasonic calibration than ninhydrin. It is based on the nucleophilic aromatic substitution reaction of trinitrobenzene sulfonic acid with primary amino groups (at the N–terminal side and lysine side chain groups), which formed the tri-nitrophenyl (TNP)–amino complex. The experimental value of $\Delta a_{r}$ obtained was equal to 0.07 ± 0.0015 kg mol^−1^ and 0.115 ± 0.0075 kg mol^−1^ for the α-chymotrypsin hydrolysis of β-lactoglobulin in 0.1-M phosphate buffer, pH 7.8 [73], and the trypsin hydrolysis of β-casein in 0.1-M phosphate buffer, pH 7, respectively, performed at 25 °C [104]. Alternatively, “Method 4” involves the computational approach using Equation (11), if changes in the volume properties, derived from the experimental methods, of the analyzed samples are known. This allows the theoretical prediction of the effects of chemical reactions on the ultrasonic velocity in solutions and could minimize and skip the requirements of the experimental calibration procedures.

The value of $\Delta a_{r}$ is expected to be constant during the reaction under the case of nonconcentrated mixtures, where there is an absence of specific interactions between the reactants and products, especially if the physicochemical properties of the products are similar to the properties of the reactants. It is also invariant within all hydrolysis of the protein as only one type of bond, –C(O)NH–, is involved in the hydrolysis and proteins having similar structural natures (e.g., compact or noncompact) as these possess small differences in the changes of hydration and the intrinsic structure [73]. This shall explain the huge difference in the Δ*a_r_* of β-lactoglobulin (compact globular) and of β-casein (noncompact micellar) presented as the example above. In addition, Δ*a_r_* may have a high dependence on pH, temperature and a medium effect, as these shall affect both the volume and compressibility properties of the molecules and the medium [73,112]. The effect of pH on Δ*a_r_* originates from the shift in the equilibrium of the ionization (mainly the proton transfer) of relevant atomic groups with respect to the pH of the medium. At a certain pH, the peptide bond hydrolysis is concerned with the partial ionization of one or more weak acids (buffer, ionizable amino acid side chains and released C- and N-terminal groups) in a solution based on the individual apparent ionization constants, $pK_{A}^{app}$ [112]. This ionization effect presents an additive contribution to $\Delta a_{r}$, and the magnitude of the contribution depends on the net change of the volume and compressibility of the ionization of participating weak acids. Following this, the net proton dissociated from a series of proton transfers takes into account the change of pH during the reaction. Thus, the evolution of terminal end groups during protein hydrolysis induces pH change. Such a change shifts the protonated and deprotonated states of each atomic group affecting the degree of the ionization reaction and, thus, the changes of the volume and compressibility of ionization. To minimize such an effect, it is ideal to perform the measurements in a buffered medium. The choice of a buffer varies with the weak acid present and responsible for the buffering action. Usually, a relatively high concentration of the buffer is being utilized as a medium of hydrolysis enough to compensate for the pH change. Therefore, the proton transfer between the buffer and the atomic groups in polypeptides dominates the ionization contribution. Likewise, the magnitude of the contribution depends on the net change of the volume and compressibility, which is a function of the pH. Finally, the temperature effect has been known to affect the hydration level through passing the energy barrier of the breakage of the hydrogen-bonded network of the hydration, resulting in more relaxed water. In addition, the dependence of $pK_{A}^{app}$ on the temperature should also be considered.

The frequency dependence measurement of the ultrasonic parameters is an effective tool to characterize the microstructure rearrangement (e.g., protein aggregation and gelation), as well as the fast kinetic reactions taken during the protein hydrolysis. As mentioned above, the proton transfer between weak acids present in the hydrolytic mixture takes place concomitant with hydrolysis. Depending on the ionization properties of the participating atomic groups, such a process gives rise to the frequency-dependent relaxation contribution to ultrasonic velocity and attenuation. This affects the amplitude of the change of the ultrasonic parameters of hydrolysis, as observed in Figure 9A [104]. In the given example, the dominant proton transfer takes place between the terminal α-amino group of protein hydrolysates and the phosphate group with a rate of constant k:(12)$$–NH_{2}+H_{2}PO_{4}^{-}\overset{k_{1}}{\underset{k_{-1}}{⇌}}\;–NH_{3}^{+}+HPO_{4}^{2-?}$$

The theoretical framework and mathematical relationship between the relaxation frequency, $f_{rel}$, was previously discussed in reference [73]. It indicates that the relaxation contribution has a positive contribution to ultrasonic attenuation but a negative contribution to ultrasonic velocity close to $f_{rel}$, usually in the region of 0.5–3 MHz. The magnitude of the frequency dependence observed is attributed to three interrelated factors: (1) the significant volume effect of the proton transfer between participating species, (2) the equilibrium concentration of the participating species and (3) the relaxation time of the process close to the relaxation frequency. The first factor is related to the thermodynamic property of the proton transfer, whereas the third factor is dependent on both the kinetics of the reaction, as well as the ultrasonic frequency of the measurements. At low-MHz frequency measurements, the comparable time scale of the oscillation of the temperature and pressure in ultrasonic waves relative to that of the relaxation process triggers perturbations of the equilibrium, giving rise to relaxation contribution. In contrast, at measurements above 10 MHz, the time scale of the oscillation of ultrasonic waves is significantly fast relative to the time scale of the proton transfer. In this case, the process is in the state of “frozen” relative to the timeframe of the ultrasonic wave oscillation, and the relaxation contribution is absent [73]. Ultrasonic velocity profiles of hydrolysis presented at a high MHz frequency do not take into account the relaxation effects but are directly attributed to changes in hydration proportional to increasing the concentration of peptide bonds hydrolyzed. Ultrasonic attenuation exhibits a better sensitivity to the relaxation process. Since the relaxation contribution is dependent on the concentration of participating species, particularly in Equation (12), ultrasonic attenuation may present a new ultrasonic methodology on the quantification of the concentration of peptide bonds hydrolyzed in a nonaggregating system.

On the other hand, the loss of the native structure and subsequent exposure of hydrophobic regions sometimes may induce a self-assembly process into a large protein aggregate structure. Their interaction with ultrasonic waves, particularly at a radius comparable to the wavelength of ultrasonic waves (including thermal and shear waves), results in a wave scattering contribution that is a function of the particle radius, frequency and volume significant at a high frequency range > 10 MHz. As aggregates may range from 10 nm up to microns size, the main scatterers are thermal and shear waves. Since a scattering contribution is likely to attenuate the ultrasonic signal, ultrasonic attenuation also exhibits a better sensitivity to the process. However, its complex dependence to the frequency and radius exhibits a bell-shaped curve, but the scattering contribution to the ultrasonic velocity is positive, exhibiting a S-shaped profile.

The real-time ultrasonic profiles of $c_{bh}$ are utilized to calculate the real-time hydrolytic profiles of the extent of the chemical reaction, ζ, and degree of hydrolysis, $d_{h}$, as well as the average degree of polymerization, $\bar{DP}$, and average molar mass of protein, $\bar{M}$, as shown below [73,99]:(13)$$\zeta =\frac{c_{bh}}{c_{P}^{0}N_{hb}};\;d_{h}=\frac{c_{bh}}{c_{P}^{0}N_{T}}$$
(14)$$\bar{DP}=\frac{{\bar{DP}}^{0}}{1+c_{bh}\frac{{\bar{DP}}^{0}-1}{c_{P}^{0}N_{T}}};\;\bar{M}=\frac{{\bar{M}}^{0}-M_{H_{2}O}}{1+c_{bh}\frac{{\bar{M}}^{0}}{w_{P}^{0}}}+M_{H_{2}O}$$
where ${C_{P}}^{0}$ is the molar (mol per kg of mixture) concentration of the protein at time zero, $N_{hb}$ and $N_{T}$ are the number of hydrolysable peptide bonds and total peptide bonds in the native protein molecule, respectively, ${\bar{DP}}^{0}$ and ${\bar{M}}^{0}$ are the average degree of polymerization and the average molar mass of the protein at time zero, respectively, $M_{H_{2}O}$ is the molar mass of water and $w_{P}^{0}$ is the weight fraction of the protein (kg in one kg of the mixture) at reaction time zero. The ratios $\frac{{\bar{DP}}^{0}-1}{c_{b}{}^{0}}$ and $\frac{{\bar{M}}^{0}}{w_{P}^{0}}$ are related to each other [73]. An example of such profiles is presented in Figure 9B and Figure 10B, calculated from the ~15-MHz data of the ultrasonic velocity profile of the protein hydrolysis.

Lynch et al. [99] also previously formulated the calculation of the change of osmolality, $\hat{m}$, using the values of ultrasonically measured $c_{bh}$ during the hydrolysis of lactose in milk:(15)$$\hat{m}=\Phi \frac{\ln(1+\bar{M}m_{s})}{\bar{M}};\;\mathrm{where}\;m_{s}=\;(m_{s}^{0}+\frac{c_{bh}}{\varphi_{w}^{0}})\frac{1}{1-\frac{c_{bh}\bar{M}}{\varphi_{w}^{0}}}$$
where Φ is the osmotic coefficient of the mixture; $m_{s}$ and $m_{s}^{0}$ are the molality of solutes at time t and zero, respectively, and $\varphi_{w}^{0}$ is the weight fraction of the water molecules mixed with solutes in unhydrolyzed media.

Finally, it is important to examine that the precision of the ultrasonic measurements of the hydrolysis curve can be calculated in the concentration of bonds hydrolyzed. The precision of our ultrasonic measurement was a maximum of 0.0002 m s^−1^, which corresponds to the approximately ±2 µmol kg^−1^ concentrations of the peptide bonds hydrolyzed, based on our values of $\Delta a_{r}$ for the proteins [104].

### 4.5. Examples of Ultrasonic Measurements of Proteolysis

The feasibility of high-resolution ultrasonic measurements for the determination of the proteolytic activity has been demonstrated and widely discussed previously. Table 1 summarizes the references on the ultrasonic monitoring of the proteolytic activity of various proteases. The earliest examples of reported applications of high-resolution ultrasonic monitoring on the protease-milk system are on the rennet-induced gelation of milk components [113,114], though an ultrasonic measurement of enzyme-induced milk coagulation using the pulse-echo technique was reported earlier [115]. The detection of the protease activity in milk during the process is not achievable using an optical assay due to the opacity of the sample. However, low-frequency rheological measurements were utilized as the main complementary methods. Chymosin is the key component of the enzyme of rennet, which is responsible for the cleavage of the peptide bonds of κ-casein and which gives rise to the aggregation of casein proteins. Using high-resolution ultrasonic measurements, multi-stage processes can be distinguished in detail from the time profile with changes in the constituent gelling particles. Dwyer et al. [113] reported that the renneting process is attributed to the increase in ultrasonic velocity caused by the hydration effect during the hydrolysis and ultrasonic wave scattering during aggregation. In Figure 11A, the different rates of the increase illustrate the multi-substage molecular rearrangements. The first slope corresponds to the increase due to the hydrolysis of κ-casein. This is followed by aggregation and gelation, which further increase the ultrasonic velocity due to the scattering effects. On the other hand, the loss of the aggregate structure concomitant with the hydrolysis slightly decreased the attenuation, and the following aggregation and gelation process increased the ultrasonic attenuation (Figure 11B). The frequency dependence observed is attributed by the ultrasonic scattering effect. Similar findings were reported by Corredig et al. [114]. The author also applied a time derivative of the ultrasonic parameters to provide an additional explanation of the processes. Wang et al. [116] utilized a single-frequency ultrasound measurement at 7.8 MHz to assess the effects of preheatment at ultra-high temperatures on enzyme-induced gelation in milks.

The potential of HR-US as a method of protease assays was also reported. A patent on the HR-US method of the screening and selection of enzymes that has the ability to produce a functionality change in milk samples and wheat flour suspension was written by Dijk et al. [117]. The action of chymosin (Maxiren^®^ 180) on milk samples and protease (Bakezyme B500^®^) on wheat flour suspension was detected through a change in the ultrasonic velocity but not kinetically measured. The analyses were rather qualitative, as no information on the enzyme reaction was reported. Niemeyer et al. [118] worked on the development of a novel ultrasonic enzyme assay for pharmacopoeia methods using HR-US. The author ultrasonically investigated the hydrolysis of the casein solution by pancreatin (mixture of enzymes) and trypsin (major component of pancreatin) under different hydrolytic conditions relevant to pharmacopoeia assays. Rather, the feasibility of the ultrasonic measurements as a potential process analytical technology (PAT) for such an online process control was examined and validated.

**Table 1 sensors-20-05594-t001:** Ultrasonic monitoring of thee proteolytic of the milk protease and milk proteins described in the literature.

Protease	Reactions and Conditions	Reference
Chymosin	Ultrasonic renneting process of milk at 30 °C. Ultrasonic analysis of multistage structural rearrangement.	[113]
Chymosin	Ultrasonic renneting process of milk at 30 °C. Comparison of effect of pH, temperature and enzyme-induced gelation in milks.	[114]
Chymosin	Ultrasonic renneting process of milk at 30 °C. Effect of preheat treatment at ultra-high temperature.	[116]
Proteinase K	Hydrolysis of Gly-Leu-Gly-Gly-Ala (synthetic pentapeptides) in 30-mM Tris buffer, pH 8, at 37 °C. Effect of substrate-enzyme ratio.	[109,119]
Proteinase K	Hydrolysis of bovine serum albumin (BSA) in 30-mM Tris buffer, pH 8, at 37 °C. Effect of enzyme concentration.	[109,119]
Proteinase K	Hydrolysis of bovine casein aggregates in 30-mM Tris buffer, pH 8, at 37 °C. Effect of substate concentration.	[109,119]
Chymosin (Maxiren^®^ 180)	Proteolytic activity in milk samples at 30 °C. Screening method of enzyme activity.	[117]
Protease (Bakezyme B500^®^) ^a^	Proteolytic activity in wheat flour suspension at 30 °C. Screening method of enzyme activity.	[117]
Trypsin	Hydrolysis of β-casein in 50-mM phosphate, pH 7.5, at 37 °C	[111]
Pancreatin	Proteolysis of casein solution at 37 °C. Effect of hydrolytic conditions relevant to pharmacopoeia assays.	[118]
α-chymotrypsin	Hydrolysis of β-lactoglobulin in 0.1-mol kg^−1^ phosphate buffer, pH 7.8, at 25 °C. Effect of substrate concentration.	[73]
Trypsin	Hydrolysis of β-casein in 0.1-mol kg^−1^ phosphate buffer. Effect of pH (6.6–8), temperature (15–45 °C) and trypsin concentration.	[104]

^a^ Neutral bacterial protease extract from *Bacillus subtilis*.

Most of protease assays were more extensively performed on a buffered matrix. The ultrasonic characterization of the proteinase K activity towards synthetic peptides, Gly-Leu-Gly-Gly-Ala, and milk proteins (BSA and β-casein micelle) in Tris buffer at pH 8 was reported by Buckin and Craig [110] and by Craig [119]. The effect of the substrate concentration was investigated, which was crucial for probing the enzyme activity and determining the Michaelis-Menten parameters. The ultrasonic measurement showed an increase in the ultrasonic velocity over time due to the loss of the intrinsic compressibility and increase in hydration. In addition, ultrasonic calibration of the ultrasonic velocity profile at 15 MHz was performed using a ninhydrin method in the hydrolysis of synthetic pentapeptides and a fluorescence method in the hydrolysis of BSA and β-casein. There was a frequency dependence in the decrease of ultrasonic attenuation observed due to a loss of bovine casein aggregates. The ultrasonic attenuation data at 15 MHz showed the highest change due to the ultrasonic wave scattering effect, which is a function of the particle radius. Born et al. [111] ultrasonically monitored the trypsin hydrolysis of β-casein in phosphate buffer using ultrasonic resonance technology (URT). The ultrasonic method is a single-frequency measurement at 10 MHz. An increase in velocity was reported and measured orthogonally with the ninhydrin assay. However, a high salt concentration was reported to limit the precision/sensitivity of the ultrasonic measurement.

The recent ultrasonic methodology, discussed in Section 4.4, has been hitherto applied on the hydrolysis of β-lactoglobulin by α-chymotrypsin [73] and β-casein by trypsin [104] in phosphate buffers. The ultrasonic profiles provide information on the evolution of the concentrations of reactants and products, which can be further derived into more advanced hydrolytic profiles, such as the average degree of polymerization and molar mass, as well as the ultrasonic osmolality. The current methodology can be extendedly applied, but not limited to, other protease systems that require a sensitivity in the buffer or even in the complex media. The sensitivity of the HR-US technique in detecting the protease was reported to be within the sub-nM range [104], which is comparable with surface-sensitive acoustic methods. This is evident from Table 2, which compares the sensitivity of surface-sensitive and HR-US methods for the detection of trypsin and plasmin. Such ultrasonic monitoring in the buffer system is an important first step required to prove the detection concepts. In addition, most protease assays have been extensively studied under a buffered matrix to unravel the intrinsic behavior of the assays.

### 4.6. Future Works and Limitations

Although the potential application of ultrasonic methodology has been described and reported, we strongly believe that there are plenty of unexplored concepts that can arise in the future. So far, the current ultrasonic methodology utilizes only one ultrasonic parameter, ultrasonic velocity, in monitoring the proteolytic activity in which its sensitivity to molecular processes, as well as data algorithms of ultrasonic calibration, have been comprehensively described. Making use of ultrasonic attenuation and multi-frequency measurements will provide the characterization of concomitant secondary reactions, such as fast chemical kinetics (proton and structural relaxation processes) and structural rearrangements (changes in the protein particle or aggregate size and phase separation).

The future applications of HR-US in the protease assay also lie in the extension of ultrasonic methodology into different protease systems. For example, it can be applied with the targeting of disease-related proteases to develop direct activity assays with clear clinical relevance. Furthermore, the growing interest for understanding protease activities in complex systems such as real biological systems or complex food matrices demand emerging methods. The ultrasonic transparency for most of the liquid or semi-solid samples makes HR-US a promising technology for such purposes. To date, only the activity of lactases in milk samples have been investigated extensively by HR-US. The ultrasonic methodology employed was rather fast and simple. Particular to the protein system, increasing the structural complexity of the analyzed samples may give rise to limitations and challenges, including the necessity of ultrasonic calibrations at different reaction conditions, clean substrate formulations, the influence of air bubbles and temperature stability, which need to be addressed. For example, the unforeseen and unwanted additional interferences may require additional studies of the system to retain the assay reliability and sensitivity. In addition, the high concentrations of other compounds present shall constitute some dependence of the calibration parameters on concentrations of other components that also require consideration.

The surface-sensitive acoustics methods are well-suited for the development of biosensors for monitoring food quality and safety. This has been already shown in the detection of plasmin in real milk samples [121]. The comparative analysis of the sensitivity of plasmin detection by QCM and ELISA methods revealed a similar sensitivity. However, while ELISA cannot detect the plasmin activity and only the whole plasmin concentration in milk, the QCM method allowed the real-time monitoring of the cleavage of β-casein at the surfaces. Thus, this method is useful for applications in dairy laboratories for monitoring the plasmin activity.

## 5. Conclusions

It has been demonstrated that surface acoustic methods, such as QCM, QCM-D, TSM and EMPAS, as well as high-resolution ultrasonic spectroscopy (HR-US), are effective tools for the detection of the activity of milk proteases. The methods are suitable for applications in nontransparent liquids, such as milk, and allowing the real-time analysis of the cleavage of proteins. The sensitivity of the surface acoustics and HR-US methods are comparable and allow the detection of trypsin and plasmin in the sub-nanomolar range, which is enough for practical applications in the milk industry, as well as in basic research focused on the study the mechanism of enzyme reactions.

Overall, we consider that HR-US may emerge as an alternative method platform for milk protease assays that provides fast, sensitive and direct sensing towards the native protein substrate in the natural or desired environment in the volume. Since proteolysis is accompanied by hydration changes, the current methodology can be extendedly applied, but not limited to, other protease systems that require sensitivity in the buffer or even in the complex media. Being a considerably valuable tool in the field of the enzyme-based industry, we also envisage that HR-US could accelerate milk protease-based research and intensively advance the field in which it will play a huge role in improving the dairy food quality.

## Figures and Tables

**Figure 1 sensors-20-05594-f001:**
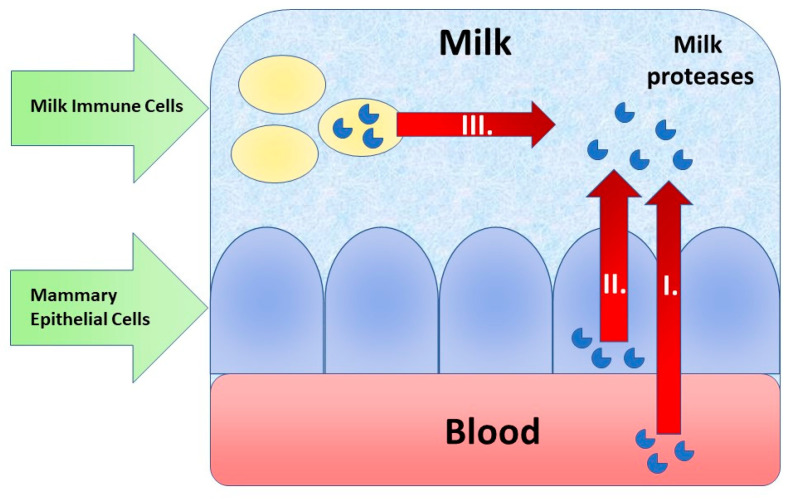
Three main routes of protease infiltration into milk. (I) Proteases from the blood pass the vessel and through the epithelial cells into milk. (II) Proteases can be secreted by mammary epithelial cells. (III) Proteases are released by milk immune cells.

**Figure 2 sensors-20-05594-f002:**
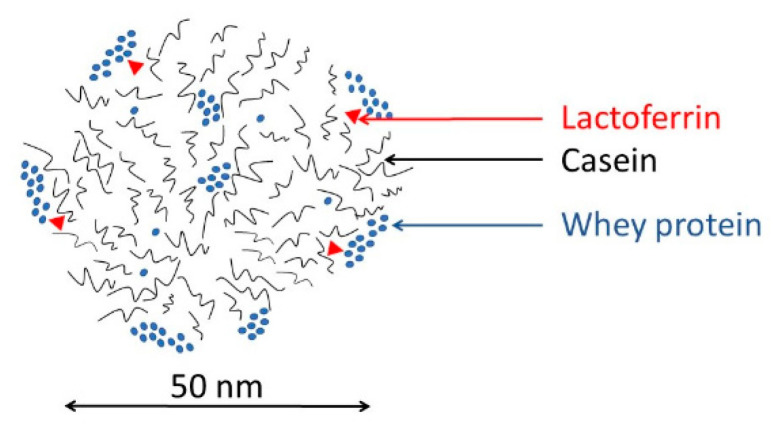
The scheme of a casein micelle. Reproduced from Halabi et al. [14] with permission of Elsevier.

**Figure 3 sensors-20-05594-f003:**
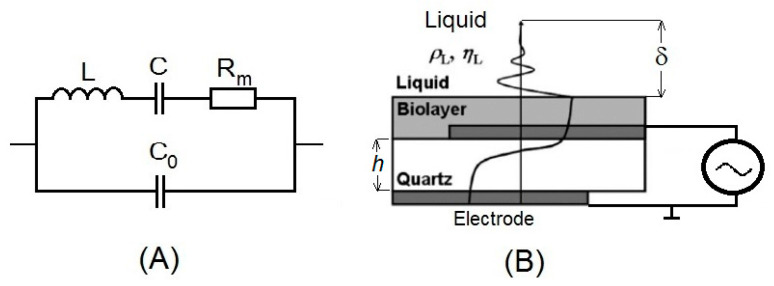
(**A**) Butterworth-van-Dyke (BvD) equivalent circuit. *C*_0_ = εA/h is the parallel electrical capacitance, *L* = 1/*ω*^2^*C* is the motional inductance (proportional to the mass), *C* = 8*K*^2^*C*_0_/(*Nπ*)^2^ is the motional capacitance (inversely proportional to the stiffness) and *R_m_* = *η_q_/μ_q_C* is the motional resistance related to the dissipative losses. *A* is the electrode area, *ε* and *h* are the dielectric permittivity and thickness of the crystal, respectively, *ω* = 2*πf*, where *f* is series resonant frequency, *K* is the electromechanical coupling coefficient and *N* is the integer. *η_q_* is the effective viscosity, and *μ_q_* is the shear stiffness. (**B**) Scheme of the propagation of the acoustic wave. *η_L_* and *ρ_L_* are the viscosity and density of the liquid, respectively. δ is the penetration depth.

**Figure 4 sensors-20-05594-f004:**
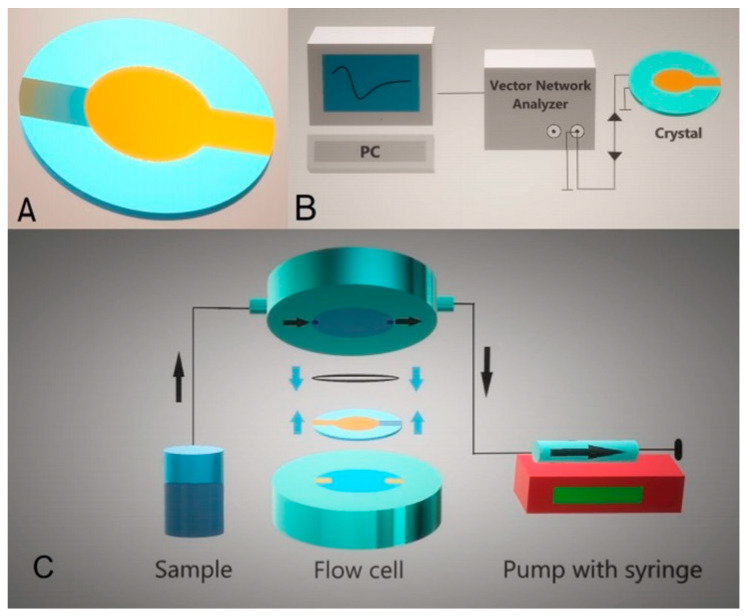
Schematics representation of the AT-cut quartz crystal with gold electrodes (**A**) and its implementation into the thickness shear mode (TSM) setting (**B**). Crystal is placed into the flow cell, and the liquid is added at the crystal surface using a syringe pump (**C**).

**Figure 5 sensors-20-05594-f005:**
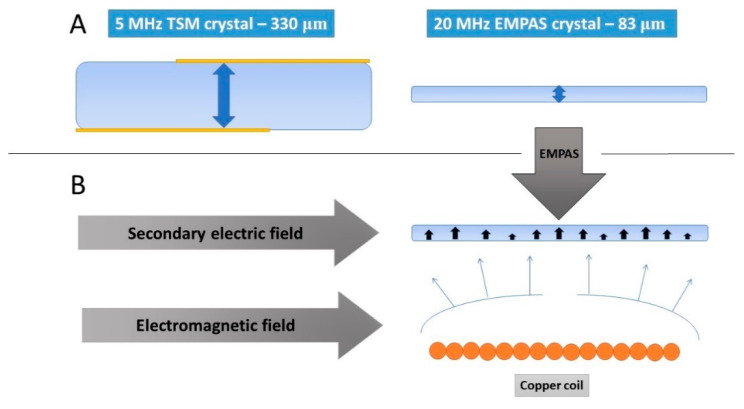
(**A**) The design and the typical thickness of the quartz crystals for the TSM and electromagnetic piezoelectric sensor (EMPAS) applications. (**B**) Schematic representation of the generation of the oscillation by the secondary electric field in the EMPAS.

**Figure 6 sensors-20-05594-f006:**
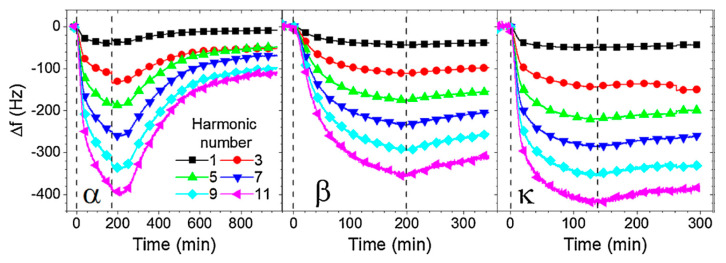
Kinetics of casein adsorption and removal from the SiO_2_ surface, as measured by a frequency shift (Δf) of the 1st–11th odd harmonics of the QCM-D. Exposure and removal of α-casein (left panel), β-casein (middle panel) and κ-casein (right panel) are marked by the first and second vertical dashed lines, respectively. Reproduced with permission of Elsevier from Tatarko et al. [62].

**Figure 7 sensors-20-05594-f007:**
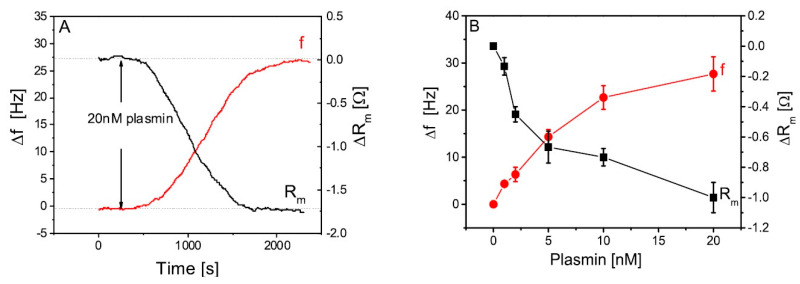
(**A**) Changes in the resonance frequency (*Δf*) and motional resistance (ΔR_m_) of the quartz crystal microbalance (QCM) transducer modified by a 1-mM peptide substrate and mercaptohexanol after the addition of 20-nM plasmin. (**B**) Plots of relative changes of the *Δf* and ΔR_m_, a function of plasmin concentration (*Δf* = *f* − *f*_0_ and ΔR_m_ = R_m_ − R_0_); here, *f*_0_ and R_0_ are the frequency and resistance measured before the addition of plasmin, respectively. The results represent the mean ± S.D. obtained from 3 independent measurements performed for each concentration of plasmin. Reproduced with permission of Elsevier from Poturnayova et al. [63].

**Figure 8 sensors-20-05594-f008:**
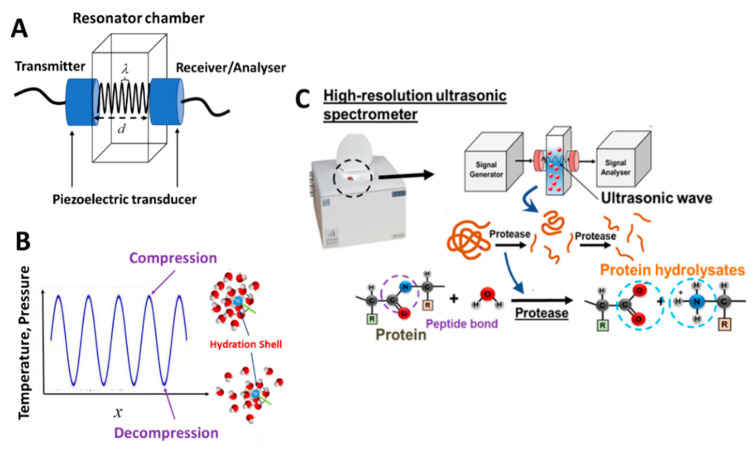
(**A**) Ultrasonic measuring system using an ultrasonic resonator system. (**B**) The ultrasonic wave propagation induces the oscillation of temperature or pressure, which results in the compression and decompression of water molecules within the hydration shell surrounding the terminal α-amino group. (**C**) Overall scheme of the peptide bond hydrolysis and high-resolution ultrasonic spectroscopy (HR-US) measuring principles.

**Figure 9 sensors-20-05594-f009:**
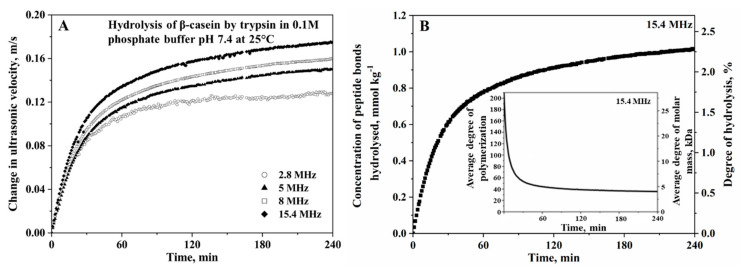
(**A**) Ultrasonic velocity profiles of the hydrolysis of β-casein treated with trypsin in 0.1-M phosphate buffer, pH 7, at 25 °C, measured at four different frequencies. (**B**) The corresponding hydrolytic profiles translated from ultrasonic velocity profiles in Figure 9A using $\Delta a_{r}$ (= 0.115 kg mol^−1^) determined from ultrasonic calibration with the trinitrobenzene sulfonic acid (TNBS) assay method. The concentration of the peptide bonds hydrolyzed (primary y-axis) is calculated using Equation (9), and the corresponding degree of hydrolysis (secondary y-axis) is calculated using the second Equation (13). The insert presents the reduction of the average degree of polymerization (primary y-axis) and average molar mass (secondary y-axis) calculated from the concentration of the peptide bonds hydrolyzed using Equation (14). The protein concentration is 5 mg mL^−1^, and the concentration of enzymes is 0.41 µM. Reproduced with permission of Elsevier from Melikishvili et al. [104].

**Figure 10 sensors-20-05594-f010:**
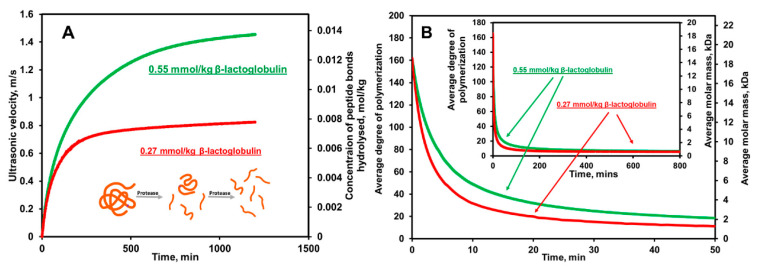
(**A**) Ultrasonic velocity profiles of the hydrolysis of β-lactoglobulin treated with α-chymotrypsin in 0.1-M phosphate buffer, pH 7.8, at 25 °C. The concentration of the peptide bonds hydrolyzed (secondary y-axis) was translated from the ultrasonic velocity profile (primary y-axis) using Equation (9) and $\Delta a_{r}$ = 0.07 ± 0.0015 kg mol^−1^. (**B**) The corresponding profile of the reduction of the average degree of polymerization (primary y-axis) and average molar mass (secondary y-axis), within 50 min of hydrolysis, calculated from the concentration of the peptide bonds hydrolyzed using Equation (14). The insert illustrates the profile at the full time range of the measurement. The concentration of enzyme is 4 µM. Adopted from Buckin and Altas [73].

**Figure 11 sensors-20-05594-f011:**
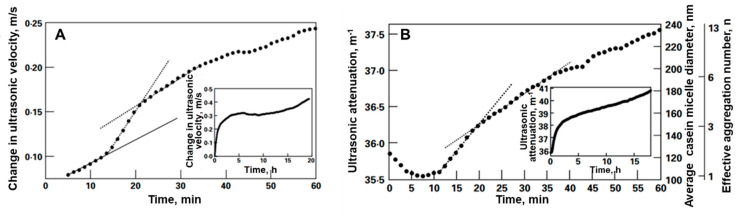
Ultrasonic profiles, measured at 14.5 MHz, of the renneting process of milk at 30 °C. (**A**) Changes in the ultrasonic velocity during the first 60 min of the renneting of the milk samples. The slopes drawn mark each region of the process. The insert presents the whole time scale of the measurements. (**B**) The corresponding ultrasonic attenuation profile showing the effects of the process. The insert presents the whole time scale of the process. Reproduced with permission of Cambridge University Press from Dwyer et al. [113].

**Table 2 sensors-20-05594-t002:** Comparison of the sensitivity of acoustics biosensors and high-resolution ultrasonic spectroscopy (HR-US) methods for milk protease detection. QCM: quartz crystal microbalance and EMPAS: electromagnetic piezoelectric sensors.

Acoustic Method	Detection Phase	Substrate	Limit of Detection	Time of Detection	Reference
QCM	surface	Short peptides at gold surface	0.65 nM Plasmin	30 min	[63]
QCM	surface	β-casein at gold substrate	0.1 nM Trypsin 1 nM Plasmin	30 min	[62]
EMPAS	surface	β-casein at gold substrate	0.032 nM Plasmin	30 min	[64]
HR-US	volume	0.1–1% (*w*/*w*) β-casein in buffer	0.2 nM Trypsin	15–30 min	[104]
HR-US	volume	0.1–1% (*w*/*w*) β-casein in buffer	0.2 nM Plasmin	15–30 min	[120]

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
