# Peer review of "Advances in Analysis of Milk Proteases Activity at Surfaces and in a Volume by Acoustic Methods"

_sensors, 2020, doi:10.3390/s20195594_

Round 1
Reviewer 1 Report
This paper reviews advances in surface and volume sensitive acoustic methods for detection milk proteases such as trypsin and plasmin. As the authors mention in the article, analysis of milk proteases activity is of research value. The authors show surface sensitive acoustic methods and high-resolution ultrasonic spectroscopy (HR-US), at the theoretical level and examples of application. I think this review is valuable, and includes enough related recent studies. However, before publication, following issues should be well addressed.
- The second part aims to emphasize how necessary this study of analysis of milk proteases activity is, but it include too much else in this section to succinctly illuminate the topic.
- Both surface sensitive acoustic methods and high-resolution ultrasonic spectroscopy (HR-US) are effective in achieving detection. Although the authors mention in the article, the sensitivity of acoustics biosensors and HR-US methods for protease detection are compared, it's a little too short about this. I think this section is extremely important and therefore suggest that the authors could expand this section with a comparative table.
- The third section of the paper is intended to illustrate the surface sensitive acoustic methods, but the beginning of the section lacks the introduction as the previous part of section 4, so it is not easy for the reader to understand.
- One of the most import parts of review paper is the authors' opinions on future / developing work in the field. However, I did not find this part in entire paper, so I would very much like the author to add these details.
- There is an error on line 453.
Author Response
Comment: This paper reviews advances in surface and volume sensitive acoustic methods for detection milk proteases such as trypsin and plasmin. As the authors mention in the article, analysis of milk proteases activity is of research value. The authors show surface sensitive acoustic methods and high-resolution ultrasonic spectroscopy (HR-US), at the theoretical level and examples of application. I think this review is valuable, and includes enough related recent studies. However, before publication, following issues should be well addressed.
Response: We are grateful to this reviewer for opinion and for very useful comments that allowed us to improve manuscript. The detailed responses to the reviewer's comments are listed below.
Comment 1: The second part aims to emphasize how necessary this study of analysis of milk proteases activity is, but it include too much else in this section to succinctly illuminate the topic.
Response: We attempted to highlight the important topics that the dairy industry is facing and that is connected with the effect of protease activity on the quality of milk and milk products. We found important to emphasize the protease system and milk proteins to introduce the reader into the problematic. Research in this topic is even broader. Many aspects are still unclear such as nonspecific inhibition of proteases and protein-protein interactions. We were therefore limited this review mostly on the principles of acoustics methods in detection proteases activity that can be implemented by dairy laboratories. In revised manuscript we included new part "4.6 Future works and limitations", in which we discussed possible future works in application of acoustics methods and a possible limitation arises in the field.
Comment 2: Both surface sensitive acoustic methods and high-resolution ultrasonic spectroscopy (HR-US) are effective in achieving detection. Although the authors mention in the article, the sensitivity of acoustics biosensors and HR-US methods for protease detection are compared, it's a little too short about this. I think this section is extremely important and therefore suggest that the authors could expand this section with a comparative table.
Response: The section has been extended and Table comparing sensitivity of surface acoustic and HR-US methods were included as Table 2 in the revised manuscript.
Comment 3: The third section of the paper is intended to illustrate the surface sensitive acoustic methods, but the beginning of the section lacks the introduction as the previous part of section 4, so it is not easy for the reader to understand.
Response: The introduction has been added at the beginning of this section as suggested by reviewer.
Comment 4: One of the most import parts of review paper is the authors' opinions on future / developing work in the field. However, I did not find this part in entire paper, so I would very much like the author to add these details.
Response: We included new part "4.6 Future works and limitations", in which we discussed future works and possible limitations, arises in the field.
Comment 5: There is an error on line 453.
Response: We are apologizing for misprint. There should be "Figure 8A". This was corrected in revised manuscript.
Reviewer 2 Report
Well written manuscript addressing a narrow topic of general interest. I would like to ask the authors to add some comments about the perspective of the proposed surface acoustic methods in routine analysis.
Author Response
Comment: Well written manuscript addressing a narrow topic of general interest. I would like to ask the authors to add some comments about the perspective of the proposed surface acoustic methods in routine analysis.
Response: We are grateful to this reviewer for positive opinion and for useful comment. We included additional text explaining perspective of the acoustic methods in routine analysis in new part "4.6 Future works and limitations" as well as in Conclusion section.
Reviewer 3 Report
The work is devoted to the detection of milk proteases by means of acoustic sensing. The review contains modern and relevant information about the topic under study. It is an interesting and useful research. However, a lot of syntax errors and punctuation mistakes were found during the review. It is desired that the manuscript is proofread thoroughly before resubmission. Specific comments follow:
Pag.1 Line 39: “… which is present in human, milk were detected”
It should be: “… which is present in human milk were detected”
Pag. 2 Line 51: “thickness shear mode acoustic method (TSM)…”
It should be: “thickness shear mode (TSM) acoustic method”.
Pag. 2 Lines 63-64: “… spectroscopy (HR-US) that allowing study of the mechanisms …”
Something sounds wrong in this sentence. Please, revise it.
Pag. 3 Lines 117-118: “… tissue-type plasminogen activator. αs2-casein and κ-casein also affecting … ”
Something sounds wrong in this sentence. Please, revise it.
Pag. 4 Lines 127-128: “… Ultra-high-temperature (UHT) processing …”
You have already defined UHT in the introduction. You should use only the acronym. Moreover, you should mention the UHT treatments.
Pag. 4 lines 148 and 154
Are the cathepsin and kallikrein systems active in both human and bovine milk?
Pag. 5 line 173: “UIPAC”
It should be “IUPAC”
Pag. 5 line 187: “stress applied to certain materials (crystals, ceramics, bone) generated electrical charges”
The stress applied on the piezoelectric material does not generate charge. It generates an electric difference potential. You should refer to the literature such as:
- Fiorillo, et al. Theory, technology and applications of piezoresistive sensors: A review. Doi: 10.1016/j.sna.2018.07.006.
- Howel. Piezoelectric energy harvesting. Doi: 10.1016/j.enconman.2009.02.020
Moreover, you should also define the piezoelectric effects (direct and converse) to understand what effect is used by the QCM for making the measurement. Refer following paper for better understanding about the piezoelectric effect in transducers:
- Fiorillo, et al. Ultrasonic Transducers Shaped in Archimedean and Fibonacci Spiral: A Comparison. Doi: 10.3390/s20102800
- Bundle, et al. Piezoelectric quartz crystal biosensors. Doi: 10.1016/s0039-9140(97)00392-5
Pag. 8 line 283: “… concertation …”
It should be “concentration”
Pag. 8 line 304: “… piesoelectric…”
It should be “piezoelectric”
Pag. 8 line 312: “… amyloid-β proteins Result correlated …”
Something sounds wrong in this sentence. Please, revise it.
Pag. 9 line 328: “… bellow”
It should be “below”
Pag. 12 line 453: “(Error! Reference source not found.A)”
What is that?
Pag. 12 line 461: “… solution and ultrasonic attenuation, α, is the energy losses in the resonance determined …”
Something sounds wrong in this sentence. Please, revise it.
Pag. 14 line 565: “… and elevated pressure.”
You should provide a reference value or range.
Pag. 14 line 577: “… varies approximately 3 ms.K-1”
You should revise the measurement unit of acoustic velocity.
Pag. 15
It seems that the size of text is different. Please, fix it.
Pag. 24 line 966: “Conclusions”
Limitation and future thoughts about the topic should be included in the conclusion part.
Author Response
Comment: The work is devoted to the detection of milk proteases by means of acoustic sensing. The review contains modern and relevant information about the topic under study. It is an interesting and useful research. However, a lot of syntax errors and punctuation mistakes were found during the review. It is desired that the manuscript is proofread thoroughly before resubmission. Specific comments follow:
Response: We are grateful to this reviewer for opinion and for very useful comments that allowed us to improve manuscript. The detailed responses to the reviewer's comments are listed below.
Comments:
Pag.1 Line 39: “… which is present in human, milk were detected”
It should be: “… which is present in human milk were detected”
Pag. 2 Line 51: “thickness shear mode acoustic method (TSM)…”
It should be: “thickness shear mode (TSM) acoustic method”.
Response: The sentences were corrected as suggested by reviewer.
Comment:
Pag. 2 Lines 63-64: “… spectroscopy (HR-US) that allowing study of the mechanisms …”
Something sounds wrong in this sentence. Please, revise it.
Response: The sentence has been revised as follows: “We will also explain principles of detection proteases activity in a volume using high-resolution ultrasonic spectroscopy (HR-US).”
Comment:
Pag. 3 Lines 117-118: “… tissue-type plasminogen activator. αs2-casein and κ-casein also affecting … ”
Something sounds wrong in this sentence. Please, revise it.
Response: This sentence was revised as follows:
“Presence of fibrin is important for the proper functionality of the tissue-type plasminogen activator, such as urokinase. The functioning of this activator is also affected by αs2-casein and κ-casein [18]”.
Comment: Pag. 4 Lines 127-128: “… Ultra-high-temperature (UHT) processing …”
You have already defined UHT in the introduction. You should use only the acronym. Moreover, you should mention the UHT treatments.
Response: We included new reference [22] on the methods of UHT treatment and modified this sentence as follows: “UHT processing of the milk must be therefore performed correctly (see Deeth et al., 1998 [22] for the methods of UHT processing of milk).”
Comment: Pag. 4 lines 148 and 154
Are the cathepsin and kallikrein systems active in both human and bovine milk?
Response: The role of cathepsin and kallikrein is described in revised manuscript as follows (Lines 149-156): “Cathepsin protease system contains several cathepsins and each of them differs in their function and presence of certain mammal species [30]. One of the cathepsin present in bovine and human milk is cathepsin D. Even when discovered in the active from, the most of the cathepsin D is in the form of inactive zymogen. Cathepsin B similarly to other proteases cleaves manly αs1-casein and β-casein. It however cleaves them at different peptide bonds. It is inactivated in basic pH and its general purpose and activity in milk is unclear. Kallikrein system has the prominent role for coagulation and fibrinolysis. Its active form is absent in milk, but some kallikrein molecules were detected in milk by proteomics [31].
Comment: Pag. 5 line 173: “UIPAC”
It should be “IUPAC”
Response: The abbreviation was corrected.
Comment: Pag. 5 line 187: “stress applied to certain materials (crystals, ceramics, bone) generated electrical charges”
The stress applied on the piezoelectric material does not generate charge. It generates an electric difference potential. You should refer to the literature such as:
* Fiorillo, et al. Theory, technology and applications of piezoresistive sensors: A review. Doi: 10.1016/j.sna.2018.07.006.
* Howel. Piezoelectric energy harvesting. Doi: 10.1016/j.enconman.2009.02.020
Moreover, you should also define the piezoelectric effects (direct and converse) to understand what effect is used by the QCM for making the measurement. Refer following paper for better understanding about the piezoelectric effect in transducers:
* Fiorillo, et al. Ultrasonic Transducers Shaped in Archimedean and Fibonacci Spiral: A Comparison. Doi: 10.3390/s20102800
* Bundle, et al. Piezoelectric quartz crystal biosensors. Doi: 10.1016/s0039-9140(97)00392-5
Response:
We included short introduction into the part 3 as follows (lines 171-177):
Piezoelectricity-based sensors are among most commonly used sensing devices on micro-scale and macro-scale level as it is evident from increasing number of scientific papers, which further broaden their application [32,33]. Piezoelectric devices are generally divided into two groups: surface acoustic wave (SAW) and bulk acoustic wave (BAW). SAW contains single-side electrodes. Generated wave deformation is defined by the crystal´s surface. While more sensitive, signal generated by devices suffers significant attenuation in biological solution. More suitable biosensors for such measurements are BAW based sensors, namely QCM and TSM [34].
According to reviewers comment we also corrected text at lines (195-198) and explained also the reverse piezoelectric effects. All references suggested by reviewer were included in the revised version:
„In 1880, Jacques and Pierre Curie demonstrated piezoelectricity. They showed that mechanical stress applied to certain materials (crystals, ceramics, bone) generated electric difference potential [38]. This direct piezoelectric effect is used even in the recent development of novel Heel Strike generators, converting mechanical energy of walking into the electrical energy [39]. In contrast, deformation caused by application of electric field is called inverse piezoelectric effect.“
Comment:
Pag. 8 line 283: “… concertation …”
It should be “concentration”
Pag. 8 line 304: “… piesoelectric…”
It should be “piezoelectric”
Response: Misprints were corrected as requested
Comment:
Pag. 8 line 312: “… amyloid-β proteins Result correlated …”
Something sounds wrong in this sentence. Please, revise it.
Response: The sentence has been corrected as follows: “They showed different effect of vitamin D2 and D3 on the oligomerization of amyloid-β proteins. Obtained results correlated with electron microscopy study.”
Comment: Pag. 9 line 328: “… bellow”
It should be “below”
Response: Misprint was corrected
Comment: Pag. 12 line 453: “(Error! Reference source not found.A)”
What is that?
Response: The misprint was removed. This should be (Figure 8A)
Comment: Pag. 12 line 461: “… solution and ultrasonic attenuation, α, is the energy losses in the resonance determined …”
Something sounds wrong in this sentence. Please, revise it.
Response: The sentence has been changed as follows: “…solution, and ultrasonic attenuation, α, which is the ultrasonic energy losses given by the bandwidth of the resonance peak.”
Comment: Pag. 14 line 565: “… and elevated pressure.”
You should provide a reference value or range.
Response: We modified the sentence as follows:
“…and elevated pressure (up to 20 bars).”
Comment: Pag. 14 line 577: “… varies approximately 3 ms.K-1”
You should revise the measurement unit of acoustic velocity.
Response. We are apologize for misprint. It should be 3 m.s-1.K-1
Comment: Pag. 15
It seems that the size of text is different. Please, fix it.
Response: It was set to Font size 10.
Comment: Pag. 24 line 966: “Conclusions”
Limitation and future thoughts about the topic should be included in the conclusion part.
Response: Additional section (4.6) on acoustic methods was added to express authors' opinions on future work and possible limitations arises in the field. However, short remarks on insights of potential HR-US and surface sensitive acoustic methods on application in other proteases system as well as its role as a non-destructive state-of-the-art sensor in dairy industry are added in the conclusion section.
Round 2
Reviewer 3 Report
Thanks for satisfying all my comments.